# Improved detection of differentially represented DNA barcodes for high-throughput clonal phenomics

Yevhen Akimov[1] (iD), Daria Bulanova[1,2], Sanna Timonen[1], Krister Wennerberg[1,2] (iD) & Tero Aittokallio[1,3,4,5,*] (iD)

## Abstract

Cellular DNA barcoding has become a popular approach to study heterogeneity of cell populations and to identify clones with differential response to cellular stimuli. However, there is a lack of reliable methods for statistical inference of differentially responding clones. Here, we used mixtures of DNA-barcoded cell pools to generate a realistic benchmark read count dataset for modelling a range of outcomes of clone-tracing experiments. By accounting for the statistical properties intrinsic to the DNA barcode read count data, we implemented an improved algorithm that results in a significantly lower false-positive rate, compared to current RNA-seq data analysis algorithms, especially when detecting differentially responding clones in experiments with strong selection pressure. Building on the reliable statistical methodology, we illustrate how multidimensional phenotypic profiling enables one to deconvolute phenotypically distinct clonal subpopulations within a cancer cell line. The mixture control dataset and our analysis results provide a foundation for benchmarking and improving algorithms for clone-tracing experiments.

**Keywords** clone tracing; DNA barcoding; fate mapping; lineage tracing; phenomics

**Subject Categories** Cancer; Methods & Resources

**Mol Syst Biol. (2020) 16: e9195**

## Introduction

Cellular DNA barcoding was originally developed to trace clonal growth dynamics *in vivo* or *in vitro* (Gerrits *et al*, 2010; Nguyen *et al*, 2014, 2015; Porter *et al*, 2014; Simons, 2016). More recently, however, cellular DNA barcoding has been applied as an effective means to detect clone-specific differences in the phenotypes other than growth, including drug response (Bhang *et al*, 2015; Hata *et al*, 2016; Lan *et al*, 2017; preprint: Acar *et al*, 2019; Bell *et al*, 2019;

Caiado *et al*, 2019; Echeverria *et al*, 2019; Merino *et al*, 2019; Seth *et al*, 2019), postsurgical recurrence (Roh *et al*, 2018), reprogramming capacity (Biddy *et al*, 2018; Shakiba *et al*, 2019), phenotypic plasticity (Lan *et al*, 2017; Mathis *et al*, 2017) and metastatic potential (Wagenblast *et al*, 2015; Echeverria *et al*, 2018; Merino *et al*, 2019). Generally, cellular DNA barcoding can be widely applied to quantify and trace in time clone-specific differences in virtually any phenotype for which a phenotype-based cell selection method exists.

Unlike single-cell RNA transcriptomics-based reconstruction of cell lineage trees from the RNA expression profiles, the DNA barcoding-based clone tracing provides an unambiguous way to trace the identity of a particular clone over time and accurately quantify the changes in the clone sizes in response to a perturbation. Therefore, emerging methodologies seek to integrate DNA barcoding-based clone tracing with single-cell technologies, such as scRNA-seq (Biddy *et al*, 2018; Fletcher *et al*, 2018; Kester & van Oudenaarden, 2018; Raj *et al*, 2018; preprint: Weinreb *et al*, 2018), or even isolate clones carrying a barcode of interest for in-depth cellular profiling (Al'Khafaji *et al*, 2018; preprint: Rebbeck *et al*, 2018; preprint: Akimov *et al*, 2019). These developments are expected to provide even more high-resolution insights into the biology of heterogeneous cellular systems. However, to our knowledge, there have been no systematic efforts to benchmark the accuracy of clonal phenotype quantification via DNA barcoding.

In a typical clone-tracing experiment (Fig 1A), cells are infected with virus particles carrying a short semi-random DNA sequence—a "barcode". The infection is performed in a very low multiplicity of infection (MOI) to ensure that each cell receives only one barcode. After that, the cells are expanded to achieve a sufficient representation of individual clones and divided into samples, typically "control" and "treatment" pools, where the control pool determines a background barcode representation, whereas the treatment pool(s) are subjected to a phenotype-based selection (e.g. drug treatment, immunophenotyping or xenografting). Finally, the barcodes are PCR-amplified from genomic DNA, and the barcode frequencies are estimated within each pool with next-generation sequencing (NGS). In the quantification phase, clone sizes are assumed to be

1 Institute for Molecular Medicine Finland (FIMM), HiLIFE, University of Helsinki, Helsinki, Finland
2 Biotech Research and Innovation Centre (BRIC) and Novo Nordisk Foundation Center for Stem Cell Biology (DanStem), University of Copenhagen, Copenhagen, Denmark
3 Department of Mathematics and Statistics, University of Turku, Turku, Finland
4 Department of Cancer Genetics, Institute for Cancer Research, Oslo University Hospital, Oslo, Norway
5 Oslo Centre for Biostatistics and Epidemiology (OCBE), Faculty of Medicine, University of Oslo, Oslo, Norway
*Corresponding author. Tel: +358 50 3182426; E-mail: tero.aittokallio@helsinki.fi

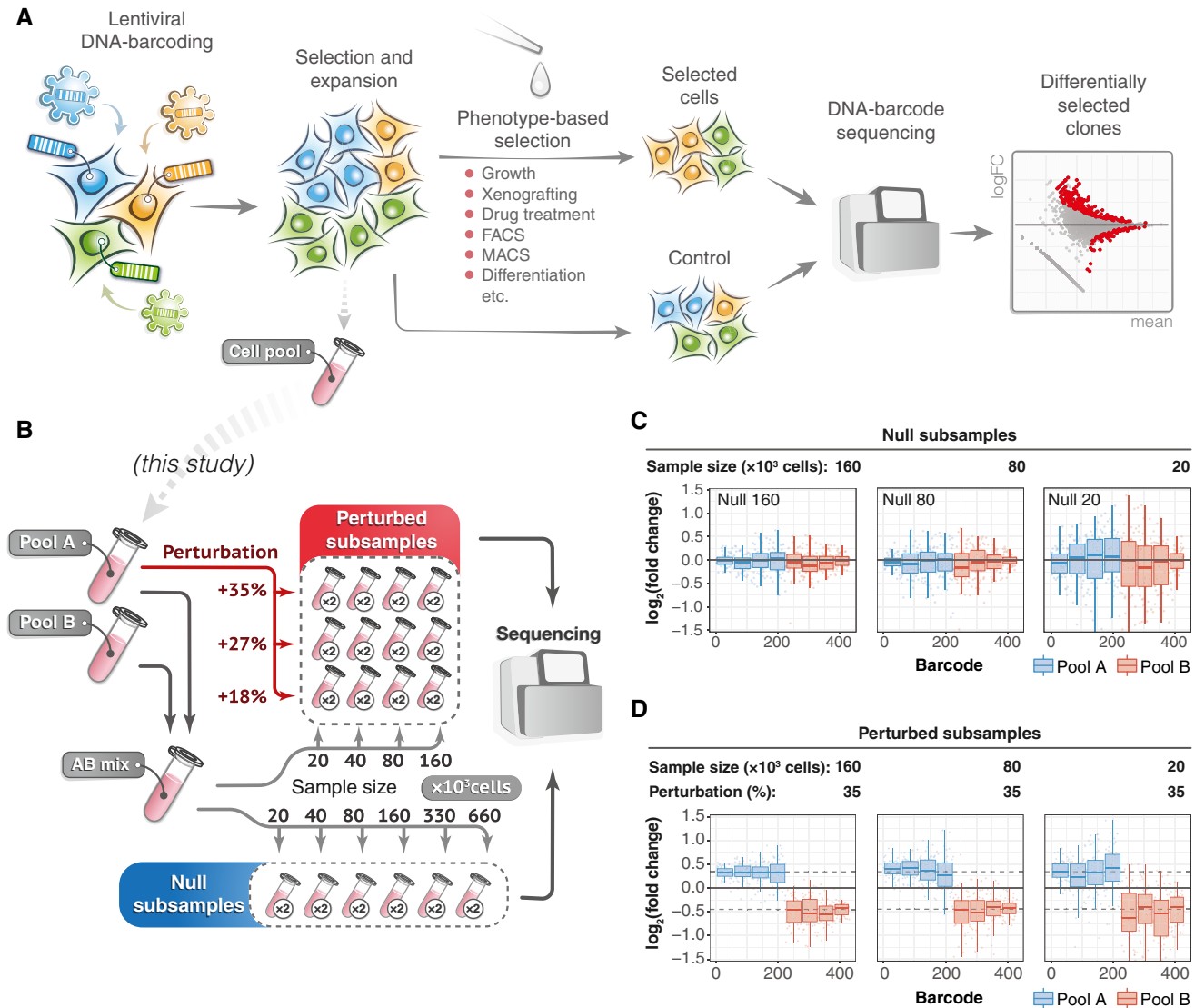

**Figure 1. An overview of the experimental setup for the benchmark dataset generation.**

A   A schematic presentation of a typical clone-tracing experiment (see text for description).

B   To generate the benchmark barcode count datasets, we performed two independent high-complexity DNA barcoding experiments on Mia-PaCa-II and OVCAR5 cell lines (see Materials and Methods for details). In each experiment, cells were collected after selection and expansion step (Fig 1A) to produce two cell pools (Pool A and Pool B). Cells in each pool were counted and mixed in a 50/50 ratio to produce "AB mix". The AB mix was then sampled in various extents in two replicas to produce so-called null samples with different numbers of cells ($20 \times 10^3$, $40 \times 10^3$, $80 \times 10^3$, $160 \times 10^3$, $330 \times 10^3$, $660 \times 10^3$), but with the same expected representation of each barcode. Perturbed samples were generated by taking either 20, 40, 80 or 160 thousand cells from the AB mix, and adding an indicated percentages of cells from the Pool A (e.g. for sample with $160 \times 10^3$ cells and perturbation degree of 35%, we added $160 \times 10^3 \times 0.35 = 56 \times 10^3$ cells from the Pool A). The number of replicas for each sample is indicated in circles next to the tube icon.

C   Barcode representation fold changes ($\log_2$) in the null samples of the indicated sizes (number of cells sampled from the AB mix), relative to the mean of two Null-660 replicas. Barcodes are ordered according to their size in the Null-660 samples. Pool A barcodes are sorted in decreasing order, and Pool B barcodes are ordered in increasing order. Boxes represent interquartile ranges for each group of 53 observations. Whiskers indicate upper and lower quartiles. Central line corresponds to the median value.

D   Same data as in (C) but for the perturbed samples. Dotted lines indicate the expected barcode fold changes calculated using formula: (cells from pool A/total number of cells)/0.5, for the Pool A barcodes, and similarly for the Pool B barcodes. Data representation is the same as in (C).

proportional to the barcode abundances, and accordingly, differentially represented barcodes (DRBs) between the treatment pool(s) and control population indicate clone-specific differences in the particular phenotype.

In statistical terms, the detection of DRBs can be considered as identification of differentially represented sequencing tags from high-throughput count data, and RNA-seq data analysis algorithms

have been applied to this task (Seth *et al*, 2019). However, we hypothesized that barcode count data from clone-tracing experiments may seriously violate the basic assumptions of the RNA-seq analysis algorithms (i.e. that tagwise variance is homogeneous and the read counts follow a negative binomial distribution). We reasoned that the tagwise variance and the underlying distribution of the barcode read counts may depend on the extent of the

sampling bottleneck introduced by the experimental manipulations (e.g. treatment). Such sample size reduction can be extremely high in some applications (e.g. high doses of a drug, cell sorting for rare subpopulations or xenotransplantation), leading to a narrow sampling bottleneck. Therefore, differences in the selection pressure (and hence sampling size) may result in a biased performance of DRB detection with the current RNA-seq analysis algorithms, unless corrected for.

Here, we performed multiple clone-tracing experiments on cancer cell lines to generate barcoded cell pools with non-overlapping sets of barcodes. We used these cell pools to produce benchmarking barcode read count datasets that model various outcomes of clone-tracing experiments. Our design simulates varying degrees of sampling-induced biases and clone-specific responses, with known ground truth to allow for benchmarking of DRB detection algorithms. We compared the commonly used RNA-seq analysis algorithms, DESeq (Anders & Huber, 2010), DESeq2 (Love *et al*, 2014) and edgeR (Robinson *et al*, 2010; McCarthy *et al*, 2012). Based on the benchmarking results, we developed DEBRA (DESeq-based Barcode Representation Analysis) algorithm for more reliable clone tracing through improved DRB detection accuracy and a proper control for false discoveries in a wide range of experimental conditions. Finally, we demonstrate how multidimensional phenotypic profiling can be implemented on barcoded cancer cells to identify phenotypically distinct clonal subpopulations.

# Results

### A benchmark dataset for modelling response heterogeneity in clone-tracing experiments

To systematically study the effect of sampling on DNA barcode count data, and the applicability of the RNA-seq data analysis algorithms to the identification of differentially responding clonal lineages, we generated a benchmark barcode read count datasets with known ground truth for differential barcode representation and realistic barcode frequency distribution. Specifically, we performed high-complexity cellular DNA barcoding experiments on two cancer cell lines—OVCAR5 and Mia-PaCa-2 (see Materials and Methods). Each cell line was independently transduced in two replicas, selected with antibiotic and expanded to produce two cell pools with non-overlapping sets of DNA barcodes (Pool A and Pool B, see Fig 1). For each cell line, the produced cell pools were mixed in a 50/50 ratio to generate the AB mix (Fig 1B), from where 18 samples of different sizes were sampled (null samples; Fig 1B). This experimental design models an experimental scenario in which different degrees of selection pressure (and hence bottleneck sizes) are applied to a sample with no clone-specific differences (Fig 1C), in response to the selection pressure (e.g. treatment). We called these samples *null samples* because no barcode is expected to be differentially represented, and therefore, an accurate DRB detection algorithm is supposed to accept the null hypothesis for the barcodes. Such null samples enabled us to study the effect of sampling size on the statistical characteristics of barcode count data and to estimate the false discovery rate of DRB detection algorithms.

Furthermore, we generated 24 *perturbed samples*, where the representation of a set of barcodes in the AB mix mixture was changed by adding extra number of cells from the barcoded cell Pool A (Fig 1B). Perturbed samples model the outcome of clone-tracing experiments on a cellular population with varying degrees of clone-specific responses to the selection pressure (e.g. treatment; Fig 1D). By sequencing of the Pool A and Pool B, we determined the ground truth for differential representation of the barcodes in the perturbed samples, which allowed us to benchmark the accuracy of the DRB detection algorithms.

### Sampling bottleneck affects statistical properties of DNA barcode count data and DRB detection accuracy

To investigate the statistical characteristics of the benchmark barcode count data, we first analysed the mean–variance relationships for each pair of null samples. We found a marked increase in the variance as the size of the sample decreases in both OVCAR5 and Mia-PaCa-2 cells (Figs 2A and B, and EV1A and B). We observed a similar dependency in the data from a pancreatic cancer patient-derived xenograft (PDX), published by Seth (Seth *et al*, 2019; Fig 2A and B), where the variance of the drug-treated samples is much higher as compared to that of the non-treated controls. The observed difference is likely due to the decrease in the total number of cells (sample size) in response to the drug treatment. We next tested how well the barcode count data follows a negative binomial (NB) distribution using the goodness-of-fit estimation for our OVCAR5 and Mia-PaCa-2 null samples and the published pancreatic PDX samples (Seth *et al*, 2019). Notably, the NB model approximated poorly the barcode count data at low count region both in the small-sized OVCAR5 null samples and in the PDX drug-treated samples (Figs 2C and EV2C). These properties of the barcode count data violate the basic assumptions made in the RNA-seq data analysis algorithms, which may lead to sub-optimal performance when applied to DRB detection in clone-tracing experiments.

To test the performance of the RNA-seq analysis algorithms for the identification of DRBs, we applied the widely used algorithms—DESeq, DESeq2 and edgeR—on the OVCAR5 null samples. An accurate DRB detection method is expected to accept the null hypothesis for all the barcodes (i.e. no barcode should be identified as differentially represented), since the representation of the barcodes is equal across the null samples. However, all the tested versions of the algorithms identified a significant number of DRBs between the null samples of different sizes, with percentages of DRBs reaching 50% at smaller sample sizes and higher FDR levels (Fig 2D). We note that all these detections are false positives, and all the algorithms had much higher type I error rates than those expected based on their empirical *P*-values (Appendix Fig S1A). DESeq performed better than the other algorithms, yet it identified more than 15% false positives at sample size of $20 \times 10^3$ cells with a nominal FDR level of 0.25. Moreover, the performance of DESeq decreased when implemented in other designs (Appendix Fig S1B). With all the tested algorithms, the proportion of falsely detected DRBs increased when comparing null samples with larger differences in size and hence variance. These analysis results show that the decrease in sample size due to the selection pressure or any other manipulation leading to cell loss may severely compromise the accuracy of DRB detection with the standard RNA-seq analysis algorithms.

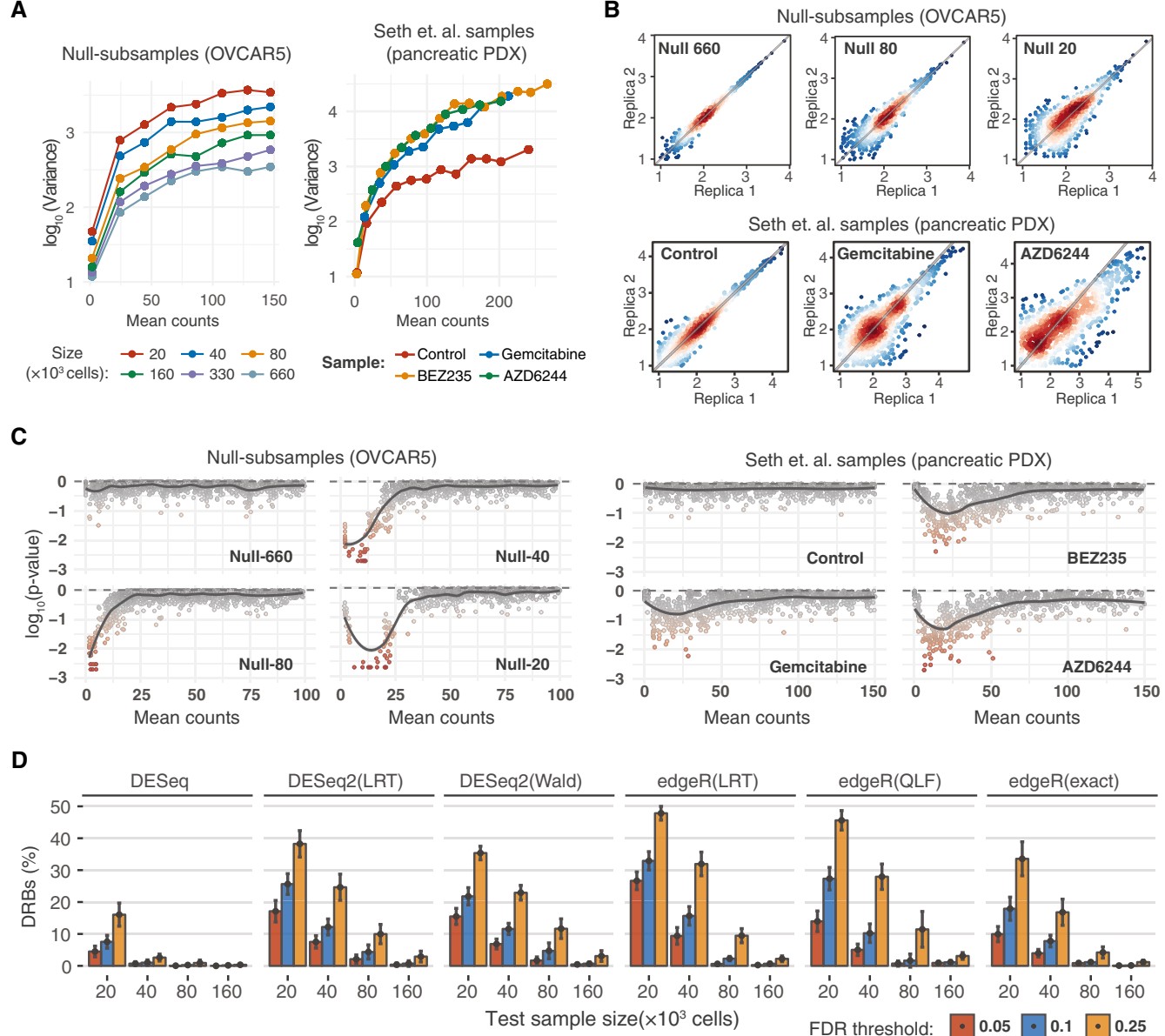

**Figure 2. Sampling size affects the statistical properties and accuracy of DRB detection.**

A Mean–variance plots for the benchmark OVCAR5 null samples and pancreatic cancer patient-derived xenograft (PDX) samples (Seth *et al*, 2019). Local variance was calculated by averaging a tagwise variance over the mean counts using a 20 read-count window.

B Scatterplots of median-normalized read counts ($\log_{10}$) of OVCAR5 null samples and pancreatic PDX samples (Seth *et al*, 2019).

C Local goodness-of-fit testing for negative binomial distribution where the distribution parameters were estimated using maximum-likelihood estimator (MLE). Two-sample Cramer–von Mises test was used to compare the observed and simulated negative binomial random variables. Statistical significance was determined using Monte Carlo bootstrap method, where a small empirical *P*-value indicates strong deviation from the negative binomial distribution.

D The proportion of differentially represented barcodes (DRBs) identified in the OVCAR5 null samples with various versions of RNA-seq analysis algorithms. Two replicas of the null samples of indicated sizes (*x*-axis) were tested for DRBs against a control group of 4 null samples (two Null-660 samples and two Null-330 samples). The bars represent the average proportion of DRBs identified with the algorithms, calculated over threefold bootstrap runs (mean of the 10 resamples with replacement) under the indicted false discovery rates (FDRs). The version with unadjusted *P*-values is shown in Appendix Fig S1A for comparison. Error bars, SD; LRT, likelihood ratio test; Wald, Wald test; QLF, quasi-likelihood *F*-test; and exact, implementation of exact test proposed by Robinson and Smyth (Robinson & Smyth, 2008), as implemented in the original algorithms.

## Modified versions of DESeq and DESeq2 algorithms effectively control for false discoveries

Mean–variance modelling is central for the inference of differentially represented tags by the RNA-seq analysis algorithms. In DESeq and DESeq2 algorithms, the tagwise variances are estimated by fitting a negative binomial (NB) generalized linear model, which assumes variance homogeneity across sample groups. However, when the variances are not homogeneous, which is the case for the DNA barcoding data (Fig 2A), the resulting tagwise estimates will become

close to the average variance between the control and treatment samples. In this case, subsequent statistical test will be performed against the NB model with dispersion parameter different from that of the treatment sample, hence compromising the accuracy of the DRB detection. Therefore, we reasoned that the observed high rates of false discoveries by standard RNA-seq analysis algorithms (Fig 2D) are caused by the differences between the variances of the control and test samples (Fig 2A). This notion is supported by the fact that the rate of false discoveries was dependent on the sample size and hence variance difference between control and treatment samples (Fig 2D). Another possible source of false discoveries is the deviance from the NB model in the low count regions (Figs 2C and EV2C), which renders the statistical tests assuming a NB model non-applicable for non-NB barcodes.

To address these statistical issues, we implemented two modifications to the DESeq2 and DESeq algorithms:

1  We modified the variance estimation procedure so that the estimation of the tagwise variances is performed exclusively from the replicates of test samples (e.g. treated samples). Two different options for the variance estimation were investigated (see below and Materials and Methods for details).
2  We implemented a heuristic algorithm that estimates a group-specific read count level (so-called β threshold, see Materials and Methods), above which the read counts follow the NB model. The estimated β threshold is used as a lower bound for the independent filtering step (Bourgon et al, 2010; Love et al, 2014).

For the variance estimation, we adopted two widely used options, which we refer to as "trended" and "shrunk" methods. The trended method corresponds to the classical approach for mean–variance relationship modelling that estimates tagwise variances from local mean–variance model as fitted by DESeq2 algorithm (via locfit R package; Loader, 2013). The shrunk method corresponds to the default method for dispersion estimation as implemented in DESeq2, where the tagwise variances are calculated via Bayesian shrinkage of individual estimates towards mean–variance trend (Love et al, 2014). The proposed β thresholding approach aims to prevent possible false discoveries arising from the read counts

which do not follow the NB model, while taking advantage of the improved detection power provided by the independent filtering algorithm (Bourgon et al, 2010; Love et al, 2014; see Materials and Methods for details). We implemented the modified DESeq and DESeq2 algorithms into a method, dubbed DEBRA (DESeq-based Barcode Representation Analysis), which is available through the Github portal (https://github.com/YevhenAkimov/DEBRA).

To benchmark the modified algorithms, we first applied DEBRA to the OVCAR5 null samples. The modified methods correctly accepted the null hypothesis for virtually all the barcodes when the null samples were tested against each other (Fig 3A), hence demonstrating a greatly improved control for false discoveries compared to the original algorithms. When the trended dispersion estimates were used, the proportion of identified DRBs were within the range of $0$–$1.5 \times 10^{-3}$, while the shrunken estimates led to somewhat increased false-positive DRB rate of up to $4 \times 10^{-3}$. To evaluate the relative contributions of the two modifications implemented in DEBRA to the false discovery rate, we first run the DEBRA algorithm without the β thresholding step. We observed a drastic drop in the number of false discoveries (Fig EV3), suggesting that incorrect dispersion estimation is responsible for most of the false discoveries. The remaining false discoveries were in the low read count region and were therefore eliminated when the β thresholding was applied (Fig EV3).

### DEBRA-modified algorithm improves the accuracy of DRB detection for various experimental outcomes

To test the accuracy of the DEBRA-modified algorithms at detecting DRBs, we used the perturbed samples to simulate experimental outcomes with varying proportions of enriched barcodes. The ground truth for the differential barcode representation in the perturbed samples was determined by assigning each barcode to the enriched or depleted group according to its presence either in Pool A or in Pool B, as defined from NGS reads of these cell pools. The ground truth information was used to generate experimental results (read count tables), with varying proportions of enriched barcodes (0.05, 0.15 and 0.5; 10 replicas for each size and perturbation degree; see Materials and Methods and Appendix Fig S2 for details). We tested each simulated experimental outcome for DRBs using the

---

**Figure 3.  Comparison of the algorithms' performance.**

Circles left to the algorithms' names indicate the modified algorithms.

A  Barplots of the percentage of DRBs identified by the modified algorithms in the OVCAR5 null samples, calculated over threefold bootstrap runs (10 resamples with replacement) using the same design as in Fig 2D. Error bars, SD.

B  The performance of the original and modified algorithms for detection of enriched barcodes in the perturbed samples. Two replicas of the sample with perturbation degree of 35%, indicated size (top) and enriched proportions (right), were tested against four null samples (two replicas of Null-660 samples and two replicas of Null-330 samples). The bars represent the average percentage of the barcodes detected as enriched DRBs (fold change > 0; FDR < 0.25) by the indicated algorithm, calculated over threefold bootstrap runs (10 resamples with replacement). Correctly assigned barcodes (classified according to the ground truth) are marked in blue and incorrectly detected barcodes (not classified according to the ground truth) are marked in red (see Fig EV3 for the results in the samples with other perturbation degrees and proportions of enriched barcodes). White circles mark the percentage of barcodes corresponding to the nominal FDR level. Error bars, SD.

C  The standardized partial area under the precision-recall curve (pAUC), calculated using intervals of [0,1] for precision and [0,X] for recall, where X is the mean recall value at FDR = 0.25 for a given sample over all the tested algorithms. The panel shows the pAUC for perturbed samples of indicted size and perturbation degree with enriched barcode proportion of 0.5 (see Appendix Figs S5 and S6 for pAUCs and precision-recall curves for other sample sizes, perturbation degrees and proportions of enriched barcodes). For calculating the precision and recall metrics, we ranked the barcodes according to their unadjusted P-values as classification scores, where the positive class was defined as correctly detected barcodes (correctly assigned to either enriched or depleted group; see Materials and Methods for details). A total of 10 threefold bootstrap runs with replacement were performed. Boxes represent interquartile ranges (25 to 75 percentile). Whiskers indicate upper and lower quartiles. Central line corresponds to the median value.

D  The full precision-recall curves for the corresponding sample sizes and perturbation degrees as in (C), with enriched barcode proportion of 0.5. For clarity, the modified algorithms with shrunken dispersion estimates are not shown here.

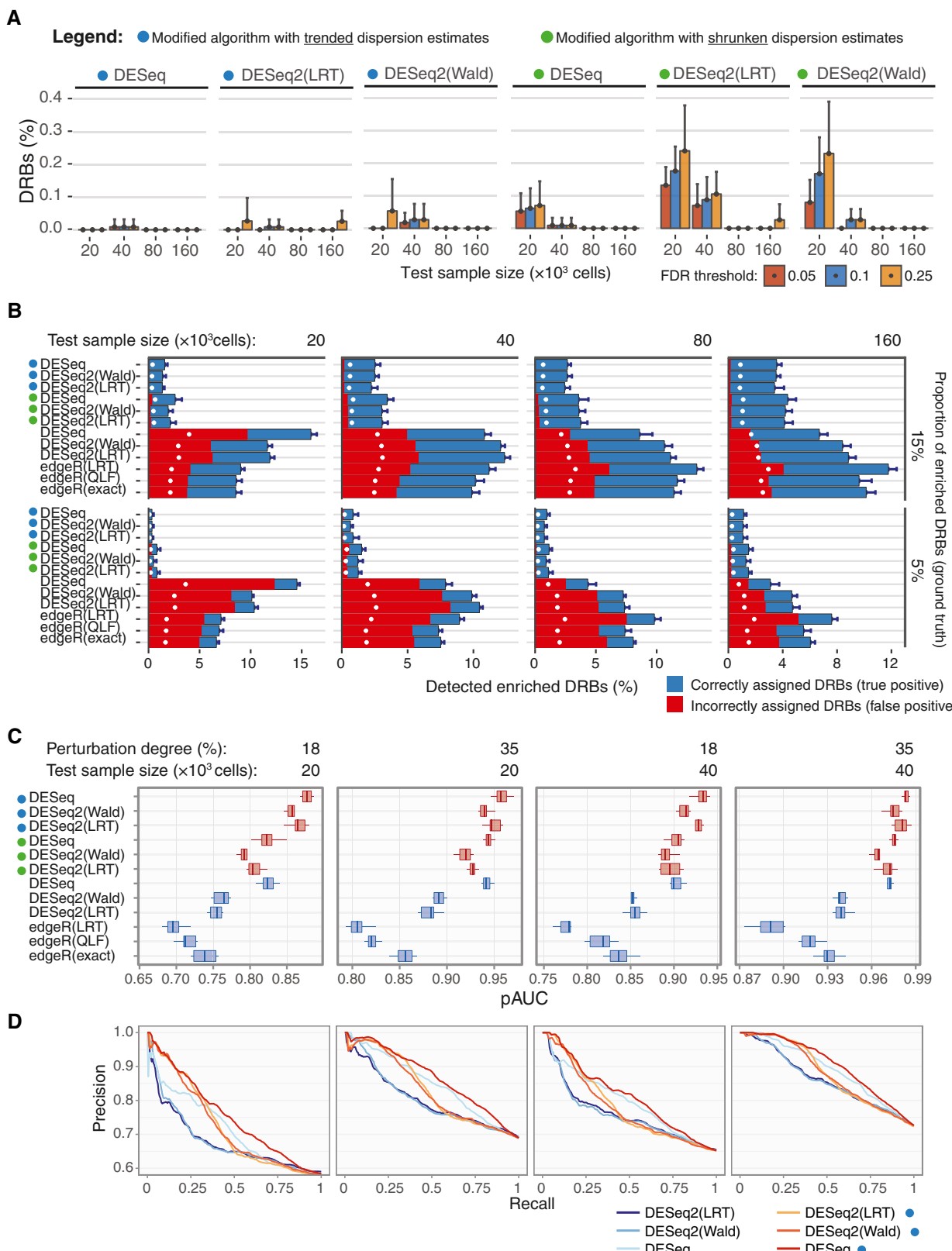

**Figure 3.**

null samples as a control. The original algorithms showed again relatively high rates of false positives in the low-size samples with enriched barcode proportions of 0.05 and 0.15 (Fig 3B). Notably, the rates of false positives were higher than expected by the nominal FDR levels (Fig 3B, white circles), except for the samples with proportion of enriched barcodes of 0.5 where the percentage of false positives dropped below the nominal FDR threshold also with the original algorithms (Fig EV3).

Notably, regardless of the proportion of enriched barcodes, the rate of false discoveries detected by the original algorithms in the low-sized samples became very close to the "random" results, that is, those obtained when the *P*-values were randomly permuted over the barcodes (indicated by black lines in Fig EV3). This suggests that the empirical significance testing implemented in the original algorithms cannot properly control for the false positives when samples with high difference in variance are compared. In contrast, the false-positive rates of the modified algorithms with trended dispersion estimates never exceeded the nominal FDR threshold, demonstrating an improved control for false discoveries when detecting DRBs across all the tested conditions (Figs 3B and EV3). To further investigate the effect of sequencing depth on the performance of the DEBRA algorithm, we down-sampled reads in the perturbed samples. We observed a robust performance of the DEBRA algorithm across various sequencing depths until reaching a critical value (40,000 reads in our dataset), after which the performance started to drop rapidly (Appendix Fig S3).

To systematically test their accuracy for DRB detection, we further compared the performance of the algorithms using partial area under precision-recall curve (pAUC) as a summary performance metric (see Materials and Methods). In this analysis, the barcodes were ranked by the unadjusted *P*-values from the algorithms, with low *P*-values indicating high statistical confidence that the barcode was either enriched or depleted. We found that the DEBRA-modified algorithms with trended dispersion estimates provided better barcode scoring in virtually all the tested scenarios, further supporting its improved performance (Fig 3C and D). Among all the tested versions, the modified DESeq with trended dispersion estimates showed the most robust scoring across all the tested conditions (Appendix Figs S3 and S4). When applied to the pancreatic PDX data (Seth *et al*, 2019), the modified algorithms with trended dispersion estimates identified again substantially less number of DRBs under the same FDR threshold than the original algorithms (Appendix Fig S7A). Consistently with results from the benchmarking dataset, the difference in the number of detected DRBs between the original and modified methods was larger for the higher-variance sample (AZD6244; Appendix Fig S7B).

## DEBRA results are consistent across bottleneck sizes in drug sensitivity experiments

Next, we sought to test whether the DEBRA-modified algorithms will improve their counterparts also in actual clone phenotyping experiments. To this end, we barcoded and expanded the OVCAR5 cells to achieve an average representation of ∼ 1,000 cells per clone, and split the cells into the control and treatment pools (Fig EV4 and Appendix Table S2). The carboplatin treatment pool was further divided into reference samples (four replicates, each with 3 M cells) and subsamples of decreasing sizes (2 × 1.33 M; 2 × 0.16 M and

2 × 0.067 M cells). Each of the treatment samples was subjected to a selection pressure—a relatively mild carboplatin treatment (IC50 of 7 μM for 4 days). The use of the mild treatment conditions as a selection pressure enabled us to produce a benchmarking dataset presenting with natural clone-specific responses, while modelling different degrees of treatment-induced sample size reduction. The reduced number of cells in the subsamples models an increasing selection pressure, while the reference samples enable calculation of FDR for the assessment of the algorithms' performance. Hence, this design allows us to benchmark the DRB detection algorithms for various degrees of bottleneck effects (i.e. sample size reduction), with realistic clone response profiles.

We first confirmed that the DEBRA-modified and original algorithms show agreement in the DRBs detected in the reference samples. Indeed, the algorithms demonstrated highly consistent FDR values in the reference samples, with DEBRA detecting slightly less DRBs at higher FDR levels (Fig EV4B). Next, we compared the FDR values between reference samples and subsamples estimated with DEBRA-modified and original algorithms (Fig 4). As expected, the larger-sized subsample (1.3 M cells) displayed a high agreement with the reference sample when analysed with either DEBRA-modified or original algorithms. However, the number of subsample-specific DRBs detected with the original algorithms (i.e. low FDR levels in the subsample but not in the reference samples) increased with the sample size reduction (Fig 4B, red frame and C, and Appendix Fig S8), which is consistent with the previously observed pattern both in the perturbed (Fig 3B) and null samples (Fig 2B). Since the decrease in the cell number in the subsamples is not supposed to increase the DRB detection power, it is likely that most of the subsample-specific DRBs are false discoveries. On the other hand, DEBRA-modified algorithms detected only a relatively few of such subsample-specific DRBs, with no observed relationship with the sample size reduction degree (Fig 4A and C, and Appendix Fig S8C and D), thereby demonstrating the desired behaviour, where the sample size reduction does not increase the proportion of false discoveries.

Interestingly, we observed a surprisingly high proportion of subsample-specific DRBs with the original DESeq algorithm, even at the largest subsample size of 1.3 M (i.e. subsample with the largest bottleneck size). We found that the majority of the DESeq-detected subsample-specific DRBs had mean count values lower than the corresponding β threshold (Appendix Fig S9). This is different from the DESeq2 detection pattern, where the number of subsample-specific DRBs below the β threshold remained relatively low in the largest subsample, and these numbers increased with the subsample size reduction degree (Appendix Fig S9). We expect that this is due to discrepancies in the dispersion estimation between the DESeq and DESeq2 algorithms. Indeed, the DESeq-derived dispersion trend was significantly lower than that of DESeq2 in the low count region. Together, these results indicate an unexpected effect of non-NB reads on the statistical inference of DRBs and stress the importance of β thresholding for reliable detection of differentially represented clones.

## Multidimensional phenotypic profiling via DNA barcoding identifies distinct cancer cell subpopulations

Clone-tracing technology has been so far primarily applied to study intrapopulation heterogeneity of cell lines and patient-derived samples. Most studies have focused on determining clone-specific

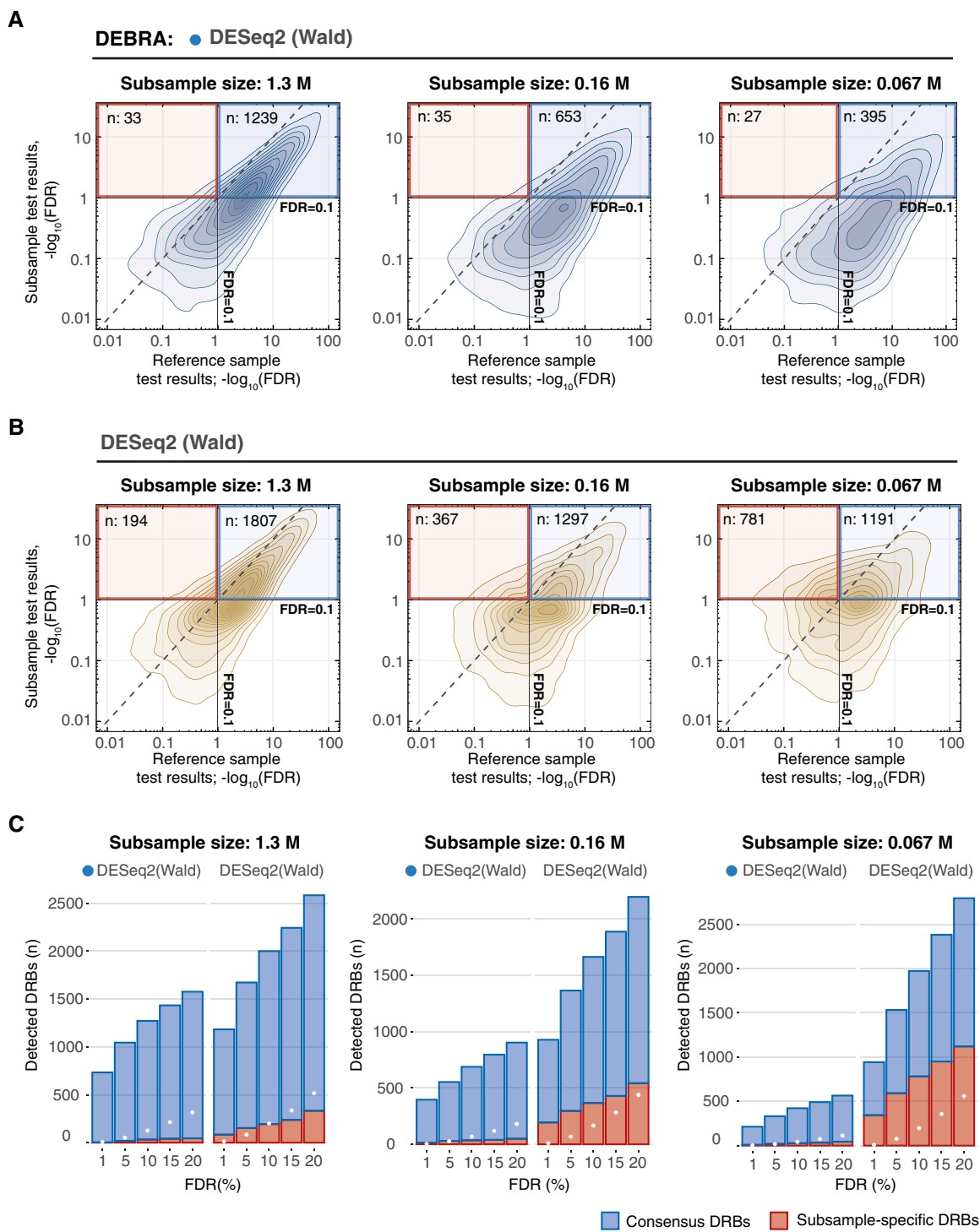

**Figure 4. Comparison of algorithm performance for carboplatin response phenotyping.**

A, B Density plots of FDR values estimated in reference sample (x-axis), plotted against corresponding FDR values in subsamples of different sizes (y-axis), where FDR was estimated with (A) DEBRA-modified DESeq2(Wald) or (B) original DESeq2(Wald). Blue square indicates the region where barcodes have FDR values lower than the set threshold of 0.1 in both reference and subsamples (consensus DRBs). Red square outlines the region where the barcodes are detected as significant only in the test subsample (subsample-specific DRBs), with the total number of detected DRBs (n) indicated in the top-left corner.

C The number of DRBs with FDR lower than a pre-selected threshold (x-axis), detected with the DEBRA-modified DESeq2(Wald) (indicated by the blue circle) and original DESeq2(Wald) algorithms in subsamples of indicated sizes (top). Blue bars indicate the number of barcodes detected as significant both in reference and in subsample (consensus DRBs), and red bars indicate the number of DRBs detected only in subsample but not in reference sample. White circles mark the percentage of barcodes corresponding to the nominal FDR level.

responses to a single perturbation, such as chemotherapeutic drugs (Bhang *et al*, 2015; Hata *et al*, 2016; Lan *et al*, 2017; preprint: Acar *et al*, 2019; Bell *et al*, 2019; Caiado *et al*, 2019; Echeverria *et al*, 2019; Merino *et al*, 2019; Seth *et al*, 2019). We argue, however, that by measuring individual responses to multiple perturbations on the same population of the barcoded cells, one can detect phenotype–phenotype associations and identify clusters of phenotypically distinct clones. Such multidimensional phenotyping depends on reliable algorithm for DRB detection and quantification. In this approach, multiple phenotypes are measured for each clonal lineage in the population to produce a multidimensional phenotypic profile. To illustrate this approach, we quantified multiple clone-specific phenotypes for the barcoded OVCAR5 cell line (Figs 5A and EV5, and Appendix Table S3) by applying a number of independent phenotypic assays that introduce differing selection pressure to barcoded OVCAR5 cells, and then analysed the clone-specific responses using DEBRA algorithm.

To confirm the capability of the approach to yield a reliable biological knowledge, we tested whether a well-known phenotypic dependence between KI67 expression and growth rate (Miller *et al*, 2018) can be detected by the DNA barcode-based phenotypic profiling (Fig 5C). Indeed, we observed a strong positive correlation between the clone-specific KI67 staining and clonal growth rates. Similarly, we found that the proliferation rate of the OVCAR5 clones correlated positively with their efflux capacity and the ability of the clones to attach to the substrate in FBS-free conditions (Fig 5D). We further tested whether these phenotypes show the same associations at the cell subpopulation level. Since the efflux and attachment assays were non-destructive to cells, we isolated populations of Efflux$^{HIGH}$, Efflux$^{LOW}$ and Attachment$^{HIGH}$ cells, and measured their growth rates. The results of these independent assays were consistent with the observed correlation at the clonal level; namely, the isolated Efflux$^{HIGH}$ and Attachment$^{HIGH}$ subpopulations showed higher proliferation rates when compared to Efflux$^{LOW}$ and bulk OVACR5 cells, respectively (Fig 5E). These results indicate that the correlation between phenotypes identified at the level of individual clones predicts phenotype–phenotype relationships at the level of cell subpopulations, suggesting the feasibility of the proposed multidimensional phenotypic profiling approach.

Next, we used the t-SNE (van der Maaten & Hinton, 2008) and UMAP (McInnes *et al*, 2018) dimensionality reduction algorithms to deconvolute cell subpopulations based on clonal phenotypes measured as fold change in barcode representation in OVCAR5 cells. We manually gated four clusters of clones with distinct phenotypic characteristics based on the t-SNE and UMAP projections accordingly (Figs 5F and G, and EV6A). Interestingly, two of the identified cell clusters displayed carboplatin resistance phenotype (Fig 5H, clusters 2 and 4). Cells from cluster 2 (∼ 8% of the population) exhibited an increased efflux capacity which is known to mediate the carboplatin resistance (Stewart, 2007; Burger *et al*, 2011). Cells from cluster 4 (about 1.5% of the population) displayed slower proliferation rates, increased ALDH activity, higher autolysosomes load and increased resistance to carboplatin (Fig 5H). Such phenotypic signature is typically attributed to cancer cells with stem-like characteristics(Ma & Allan, 2011; Vitale *et al*, 2015; Tomita *et al*, 2016; Peng *et al*, 2017; Sharif *et al*, 2017; Boya *et al*, 2018; Nazio *et al*, 2019). Within the largest cluster of clones (cluster 1), sensitivity to carboplatin showed a moderate correlation with proliferation rate (Fig EV6B).

Taken together, the high-throughput phenotyping of the barcoded clones suggests that the OVCAR5 cell resistance to carboplatin could emerge through various mechanisms mediated by different cell subpopulations. Furthermore, these data suggest that the high-throughput phenomics approach via DNA barcoding enables inference of phenotypically distinct clonal subpopulations, even within a cell line. To test whether the modified significance testing implemented in the DEBRA algorithm provides an advantage also for the clone clustering analysis, we compared the UMAP projections of clones selected by FDR value from DEBRA-modified DESeq2(Wald) and those from the original DESeq2 (Wald) algorithm. The clustering of DEBRA-selected clones

---

**Figure 5. Multidimensional phenotypic profiling approach.**

A   A schematic presentation of the experimental workflow for barcoding-based high-throughput multidimensional clonal lineage phenomics approach. Cells were barcoded and expanded to achieve reasonable representation of cells per barcode (e.g. 500–4,000). Next, the population was divided into multiple samples and selection pressure was applied to each sample. Cells passing selection conditions were collected and used to prepare a NGS library. In the present study, we measured clone-specific fold changes in barcode representation in the following assays: carboplatin response (7 μM carboplatin for 3 days followed by 7 days re-growth), autophagy measured by autolysosomes load (FACS; Thomé *et al*, 2016), ALDH activity, activity of efflux pumps, proliferation (7 days), 12 h of attachment assay in FBS-free media (attached and non-attached cells were collected) and KI67 staining.

B   Barcode representation fold changes in response to the indicated selection pressure for the clones with mean normalized read counts larger than 70.

C   Scatterplot of fold change in the barcode representation after 7 days of growth versus fold change in representation between KI67$^{HIGH}$ population and control. Each point represents a clone with colour indicating the local density of points. Displayed are only clones with counts larger than 70. *R*, Pearson correlation coefficient.

D   Scatterplot of fold changes in clone abundances after attachment in FBS-free condition and 7 days of growth (left), or upon sorting by efficacy of fluorescent dye efflux and 7 days of growth (right), as described in Materials and Methods.

E   The average doubling time of the cell subpopulations separated by their attachment to substrate in FBS-free conditions (left; *n* = 6 biological replicates for each group) or sorted by their efficacy to efflux fluorescent dye (right; *n* = 3 biological replicates for Efflux$^{HIGH}$ and *n* = 6 biological replicates for Efflux$^{LOW}$). *P*-values are from Wilcoxon test. Boxes represent interquartile ranges. Whiskers indicate upper and lower quartiles. Central line corresponds to the median value.

F   t-SNE and UMAP projections of the OVCAR5 clonal phenotypic profiles, each point represents a clone coloured according to the manually gated clusters based on the t-SNE projection. The clones with read counts of more than 70 were used for the statistical analysis.

G   UMAP projections of the OVCAR5 clonal phenotypic profiles. The clones are colour-coded according to the manifestation of the phenotype, calculated as log$_2$ ratio of barcode fractions between (1) positively and negatively selected populations after ALDH, attachment, efflux capacity or autophagy assays; (2) treated and untreated samples for carboplatin treatment assay; or (3) day 8 and day 1 time points for proliferation assay.

H   The distribution of log$_2$ fold changes by clusters (*n* = 201, 19, 31, 7 for clusters 1–4, correspondingly) in barcode representations upon selection for the indicated phenotypes. *P < 0.05; **P < 0.01; ***P < 0.001; ****P < 0.0001; ns, non-significant, based on Wilcoxon test. Boxes represent interquartile ranges. Whiskers indicate upper and lower quartiles. Central line corresponds to the median value.

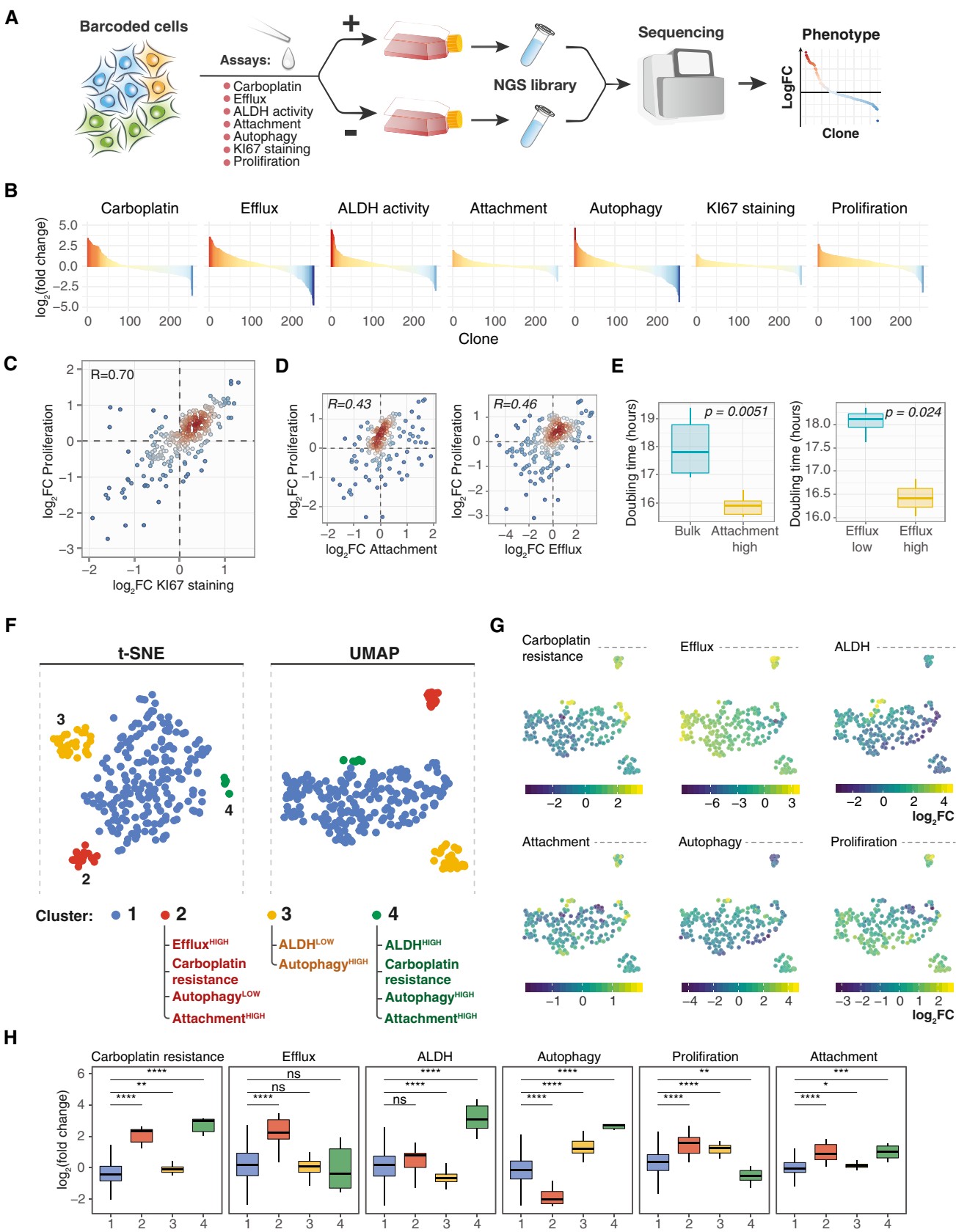

**Figure 5.**

produced visually more distinguishable clusters, and also better recapitulated the previously detected cell clusters (Fig EV6C and D). This suggests that DEBRA algorithm benefits the downstream analyses of DNA barcoding experiments, including multidimensional phenotypic profiling and identification of cells for subsequent profiling experiments.

# Discussion

Clone tracing via DNA barcoding is a promising method for the studies of intrapopulational heterogeneity of cellular systems. The method is already well-established for tracing growth dynamics of single clones, but the recent studies suggest a much wider scope of the method's applicability. In essence, these studies utilize clone-tracing technology to detect clone-specific differences for a phenotype of interest, which are often associated with a strong selection pressure that imposes a narrow sampling bottleneck and reduced cell numbers, and hence, decreased barcode representation. Here, we demonstrated that the sampling bottleneck affects statistical properties of the barcoding read count data, hence influencing the detection accuracy of differentially represented tags. By accounting for these statistical properties, we implemented an algorithm (named DEBRA), for reliable detection of DRBs, and demonstrated through systematic benchmarking against the state-of-the-art RNA-seq data analysis algorithms that DEBRA improves the accuracy of detection of differentially responding barcodes in various experimental conditions. The mixture control dataset and our analysis results provide a systematic foundation for benchmarking and improving future algorithms for DNA barcoding data analysis.

In the clone-tracing experiments, the physical number of individual sequencing tags (barcodes) in the sample is close to the total number of cells, whereas in RNA-seq experiments, each cell has around $10^5$ RNA molecules resulting in a much higher number of individual tags in the sample. Hence, we reasoned that in a clone-tracing experiment, unlike RNA sequencing experiment, a decrease in the cell numbers associated with treatment procedures could impose a strong sampling error on barcode representation in a manner dependent on the degree of the sample size reduction. Our results supported this notion, as we observed strong dependency between sample variance and sample size (Fig 2). It is also tempting to speculate that the observed deviance from negative binomial model at low count region is caused by large values of sampling error for barcodes with low copy number (physical number of DNA barcodes in the genomic DNA preparation). Although we prepared the sequencing libraries right after subsampling, we expect that the variation imposed by the sampling bottleneck is preserved also when the samples are allowed to re-grow, something that may have happened in the Seth *et al* experiments. We note that increasing the cell expansion times to achieve higher clone abundances is not a straightforward solution for the sampling issue. In fact, the expansion time is an indispensable experimental parameter of a clone-tracing experiment, as clonal phenotypes are subject to change as a result of phenotypic plasticity (Gupta *et al*, 2011; Porter *et al*, 2014), which may dilute phenotypes determined by non-genetic factors (e.g. epigenetics). Hence, limiting the expansion times is expected to improve quantification of clonal phenotypes. Therefore, there is a critical need for accurate detection of DRBs especially in

samples with low clone abundances, using DEBRA or similar algorithmic solutions.

To benchmark the performance of the original and modified algorithms for DRB identification, we simulated clone-tracing experiments with rather challenging scenarios. In the benchmarking mixture cell pool experiments, we therefore used a relatively low number of cells per barcode together with low effects sizes (perturbation degrees of 18, 27 and 35%). These experimental setups are not merely simulated scenarios; in fact, many applications of clone tracing are carried out in the context of a very strong selection pressure (e.g. exposure to high doses of drug, xenografting or cell sorting for rare subpopulations), leading to a narrow sampling bottleneck. The DEBRA algorithm was able to both prevent an excess of false discoveries, as assessed with the null and perturbed samples, and improve the accuracy of DRB classification, compared to the original algorithms, as evaluated with precision-recall analysis (Fig 3). The results from the perturbed samples should be interpreted relatively to expected "random" FDRs (indicated by black line in Fig EV3). For instance, the relatively low proportion of false discoveries observed in the samples with 50% of enriched barcodes (Fig EV3C) does not indicate a proper statistical inference, since the level of false discoveries is very close to the "random" FDR.

The DEBRA R package also provides the user with a functionality to choose between two dispersion estimation algorithms—"shrinkage" and "trended". The "trended" method assumes a strict relationship between means and dispersions, whereas the "shrinkage" uses dispersions as estimated by the DESeq2 algorithm. DESeq2 shrinks tagwise dispersion estimates towards dispersion trend using an empirical Bayes approach while allowing for dispersion outliers (Love *et al*, 2014). In RNA-seq experiments, this helps to deal with genes whose dispersions do not strictly depend on the mean and therefore cannot be approximated merely by the dispersion trend. The dispersion outliers are typically attributed to either technical or biological factors. However, it is not clear whether these effects arise also in the clone-tracing experiments. We observed a more robust performance of the trended method for dispersion estimation, which may be attributed to the absence of such factors in DNA barcoding experiments. However, we reasoned that the dispersion outliers could still appear in other clone-tracing experiments with strong selection pressure followed by the long re-growth phase. For instance, some smaller-sized clones experiencing higher sampling error during the selection phase could then re-grow to a larger relative size (shifting to a greater mean count values), thereby becoming as dispersion outliers. Further studies are needed to better understand the relative benefits of the different dispersion estimation methods in various experimental setups.

We expect that the development of the DNA barcode-based clone-tracing approach for phenotyping and the implementation of the reliable detection algorithms proposed in the current study will significantly aid the adoption of the technology for a broader spectrum of applications, such as clonal mechanisms of tumour initiation, immune evasion, metastasis, differentiation and tissue regeneration. Specifically, the increased specificity of the DEBRA algorithm enables one to faithfully test for differential phenotypes of smaller-sized clones and at lower effect sizes, hence providing high-resolution clonal phenotypic profiles. Accurate detection of clones

with differential phenotypes is also critical for the validity for the follow-up experiments and subsequent biological conclusions. The DEBRA approach could also become useful in the analysis of positive selection CRISPR screens, where the selection pressure is applied to the screening pool (cells expressing Cas9 and sgRNA library), and the representation of sgRNAs in treatment pool is compared to the background distribution. Similar to the DNA barcodes, the sgRNAs may undergo significant representation bottleneck depending on the degree of the selection pressure. Therefore, appropriate control for the variance differences between control and treatment samples, as implemented in DEBRA, may be required for accurate inference of the differentially represented sgRNAs in positive selection CRISPR screens, thereby complementing the repertoire of the pooled screening data analysis methods (Li *et al*, 2014; Hart & Moffat, 2016).

Finally, we introduced a DNA barcoding-based clone phenomics approach, which links multidimensional phenotypes to clones in a high-throughput manner (Fig 5). We expect this approach will expedite the inference of cellular subpopulations with distinct phenotypic properties, finding associations between multiple phenotypes and to improve the quantification accuracies when analysing intrapopulational phenotypic heterogeneity. The obtained information on the single-clone phenotypic state could be further integrated with single-cell technologies. For instance, simultaneous readouts of expressed DNA barcodes and single-cell gene expression via scRNA sequencing enable single-cell trajectory tracing (Biddy *et al*, 2018; preprint: Weinreb *et al*, 2018). In such combination, the readout via scRNA-seq provides both single-cell expression profiles and clone identities. The clone identities could then be used to link single-cell gene expression to the multidimensional phenotypic profiles of clonal lineages. Similar integration of clonal lineage phenomics with other single-cell approaches, e.g., single-cell genotyping or scATACseq (Navin *et al*, 2011; Cusanovich *et al*, 2015; Guo *et al*, 2017; Kim *et al*, 2018), could also promote the discovery of genetic and non-genetic determinants of intrapopulation phenotypic heterogeneity in tumours, given a reliable testing for differentially represented clones enabled by the DEBRA algorithm.

# Materials and Methods

### Reagents and Tools table

| Reagents/Resources | Reference or source | Identifier or catalog number |
|---|---|---|
| **Experimental models** | | |
| OVCAR-5 | NCI collection | – |
| Mia-PaCa-2 | ATCC | CRL-1420 |
| HEK 293FT | Thermo Fisher Scientific | R70007 |
| **Recombinant DNA** | | |
| B-GLI-Barcoding | This study | |
| pCMV-dR8.2 dvpr | Addgene | #8455 |
| pCMV-VSV-G | Addgene | #8454 |
| **Reagents** | | |
| Lipofectamine 2000 | Thermo Fisher Scientific | 11668019 |
| NucleoSpin® Tissue kit | MACHEREY-NAGEL | 740952.50 |
| Platinum SuperFi II DNA Polymerase | Thermo Fisher Scientific | 12361010 |
| AMPure XP SPRI beads | Beckman Coulter | A63880 |
| OneTaq® DNA Polymerase | New England Biolabs | M0480 |
| AarI restriction enzyme | Thermo Fisher Scientific | ER1581 |
| Plasmid-Safe™ DNase | Lucigen | E3101K |
| Endura™ *Escherichia coli* | Lucigen | 60242-2 |
| NucleoBond® Xtra Midi Kit | MACHEREY-NAGEL | 740410.50 |
| NucleoSpin® Gel and PCR Clean-up kit | MACHEREY-NAGEL | 740609.50 |
| NEBNext® Ultra™ II Q5® Master Mix | New England Biolabs | M0544 |
| Rapid Ligation Buffer | Thermo Fisher Scientific | K1422 |
| T4 DNA Ligase (5 U/µl) | Thermo Fisher Scientific | EL0014 |
| Carboplatin | MedChemExpress | HY-17393 |
| Aldefluor ALDH activity Kit | STEMCELL Technologies | 01700 |
| Rabbit anti-Ki67 antibody | Abcam | 16667 |

**Reagents and Tools table** (continued)

| Reagents/Resources | Reference or source | Identifier or catalog number |
|---|---|---|
| Goat anti-Rabbit Secondary Antibody, Alexa Fluor 555 conjugated | Thermo Fischer Scientific | A27039 |
| Acridine orange hemi(zinc chloride) salt | Sigma-Aldrich | A6014 |
| CDy1 fluorescent dye | Active Motif | 895 |
| Tariquidar | Selleck | S8028 |
| Probenecid | Santa Cruz Biotechnology | sc-202773 |
| SH800Z Cell Sorter | SONY | – |
| Cytation 5 | BIOTEK | – |
| **Oligonucleotides** | | |
| Barc.LGMU6.templ (ssDNA) | This study | Table EV1 |
| Barc.LGMU6.aarI.ampl.F | This study | Table EV1 |
| Barc.LGMU6.aarI.ampl.R | This study | Table EV1 |
| B-GLI_Barcoding (9,301 bp) | This study | Table EV1 |
| P5.seq-B-GLI.v1 | This study | Table EV1 |
| P7.seq-B-GLI.v1 | This study | Table EV1 |
| Illumina_indX_F | This study | Table EV1 |
| Illumina_indX_R | This study | Table EV1 |

## Methods and Protocols

### Generation of the lentiviral plasmid barcode library

Semi-random single-stranded DNA template (Barc.LGMU6.templ; Table EV1) from Merck (Sigma-Aldrich) was used in the work. The oligonucleotide was amplified with Barc.LGMU6.aarI.ampl.F and Barc.LGMU6.aarI.ampl.R primers (Table EV1), using SuperFI DNA polymerase to include cloning overhangs compatible with Golden Gate cloning. Five microlitres of the reaction was transferred to a new 50 μl PCR with an excess of Barc.LGMU6.aarI.ampl.F and Barc.LGMU6.aarI.ampl.R primers (Table EV1). The reaction was run one cycle (2 min at 98°C denaturation, 5 min 72°C annealing/ elongation) to produce dsDNA barcodes with no mismatches. The barcode cassette was purified with AMPure XP SPRI beads (Beckman Coulter; catalog number A63880). The barcode cassette was then cloned into previously generated B-GLI-Barcoding plasmid (preprint: Akimov et al, 2019; see Appendix Fig S10 for the plasmid map and Table EV1 for the DNA sequence) by the Golden Gate assembly method (Engler et al, 2008; see Appendix Table S4 for reaction composition and cycling conditions). In order to reduce contamination with uncut B-GLI-Barcoding plasmid, an extra 2 μl of the AarI enzyme was added to the reaction after the Golden Gate cycling, followed by incubation at 37°C for 16 h. The cloning reaction was purified with magnetic beads (Beckman Coulter; catalog number A63880) and incubated with Plasmid-Safe™ DNase (Lucigen, catalog number E3101K), according to the manufacturer's instructions. The reaction was again magnetic bead-purified and transformed into electrocompetent Lucigen Endura™ E. coli (Lucigen; catalog number 60242-2) using Bio-Rad MicroPulser Electroporator (catalog number #1652100) with program EC1 following the manufacturer's instructions. The reaction was plated onto 5 × 15 cm LB-agar plates with 100 μg/ml ampicillin. After incubation for 16 h at 32°C, bacteria were collected and plasmid DNA was extracted with NucleoBond® Xtra Midi Kit (MACHEREY-NAGEL; catalog number 740410.50). The efficiency of transformation and approximate number of the unique barcodes in the library was assessed by plating 1/10,000 of the reaction onto 15-cm LB-agar plate with 100 μg/ml ampicillin and counting colonies after overnight incubation at 37°C.

### Lentivirus packaging

HEK 293FT cells were seeded at a density of $10^5$ cells per cm$^2$. Next day, the cells were transfected with a transfer plasmid, packaging plasmids pCMV-VSV-G (Stewart, 2003; Addgene plasmid #8454) and pCMV-dR8.2 dvpr (Stewart, 2003) using Lipofectamine 2000 Transfection Reagent according to the manufacturer's instructions. Virus supernatants were collected 48 h post-transfection. The titre of the virus was determined as described (Stewart, 2003; Najm et al, 2018).

### Generation of null and perturbed samples

OVCAR5 and Mia-PaCa-2 cells were seeded at a density of $2 \times 10^4$ cells/cm$^2$ and $1 \times 10^5$ cells/cm$^2$, respectively, both in 6-well plates in two replicas. Cells were incubated overnight with lentiviral barcoding library carrying ~ $5 \times 10^6$ unique barcodes in a presence of 8 mg/ml polybrene. The amount of added virus was selected to achieve a multiplicity of infection (MOI) of ~ 0.01. Cells were selected for 7 days in the presence of 150 μg/ml hygromycin. Cells were kept at a density of at least $1 \times 10^4$ cells/cm$^2$ to improve viability during selection and expansion. Cells were expanded to achieve approximately 4,000 cells per clone (12 cell divisions) to produce two cell pools for each cell line (Pool A and Pool B). Cells from each pool were counted and mixed in a 50/50 ratio to produce the AB mix (Fig 1B). The AB mix was then subsampled to various extents in two replicas to produce null samples with different sizes ($20 \times 10^3$, $40 \times 10^3$, $80 \times 10^3$, $160 \times 10^3$, $330 \times 10^3$, $660 \times 10^3$ cells) but with the same expected representation of each barcode (i.e. modelling null hypothesis).

Perturbed samples were generated by taking either 20, 40, 80 or 160 thousand cells from the AB mix, and perturbing it by adding extra number of cells from Pool A to achieve the intended perturbation degree (Fig 1B). For example, to achieve a perturbation degree of 35% for a sample with $160 \times 10^3$ cells, we added $160 \times 10^3 \times 0.35 = 56 \times 10^3$ cells from the Pool A.

### NGS library preparation and sequencing

NucleoSpin® Tissue Kit (MACHEREY-NAGEL) was used to isolate genomic DNA according to the manufacturer's instructions. Barcodes were amplified from genomic DNA with P5.seq-B-GLI.v1 and P7.seq-B-GLI.v1 primers using OneTaq® DNA Polymerase (NEB; catalog number M0480). Reactions were purified using NucleoSpin® Gel and PCR Clean-up Kit (MACHEREY-NAGEL). Then, purified amplicons were amplified with primers, Illumina_indX_F and Illumina_indX_R (where X indicates the index sequence), to add Illumina adapters and indexes for sample multiplexing. This round of PCRs was performed using NEBNext® Ultra™ II Q5® Master Mix (NEB, catalog number M0544). Samples were purified using AMPure XP beads (Beckman Coulter; catalog number A63880). Next-generation sequencing library was sequenced with HiSeq 2500 Illumina sequencer using 100-bp paired-end protocol (with 10% PhiX DNA spike-in). To improve cluster calling, we increased sequence diversity by using a 15-bp random sequence stagger in the P5.seq-B-GLI.v1 primer.

### Barcode retrieval from NGS data

We used the previously developed (preprint: Akimov *et al*, 2019) custom Python script for retrieving original barcode counts from FASTQ files.

### Generation of experimental datasets with varying proportions of enriched clones

Using the perturbed samples, we generated datasets with varying percentage of the enriched barcodes. For this experiment, we assigned a ground truth for every barcode based on sequencing results of Pool A and Pool B samples (Fig 1B). Then, we sampled barcodes from the read count dataset of a perturbed sample of interest to generate a simulated dataset with the required proportion of the enriched barcodes (Appendix Fig S2). For instance, to generate a dataset with 5% enriched barcodes, perturbation degree of 35% and size of $20 \times 10^3$ cells, we randomly sampled barcodes detected in the Pool A (enriched in the perturbed samples) and Pool B (depleted in the perturbed samples) in the 5/95 ratio from the columns of the read count table corresponding to the perturbed samples with 35% perturbation degree and $20 \times 10^3$ cells (m_null_20.p35.1; m_null_20.p35.2; see Appendix Table S1).

### Running DESeq, DESeq2 and edgeR

Dispersion estimation in DESeq (Anders & Huber, 2010) and DESeq2 (Love *et al*, 2014) algorithms was implemented using *fitType = "local"* parameter, as the *"parametric"* fit option resulted in frequent errors, possibly due to the statistical properties of the barcode count data. Furthermore, we used *method = "per-condition"* setting in DESeq algorithm. The in-built independent filtering option was switched off in DESeq2. The edgeR algorithm was run with its default parameters (Robinson *et al*, 2010). We used " ~ *condition*" formula for finding differentially represented barcodes between control and treatment groups.

### DEBRA implementation aspects

#### The β threshold estimation

The DEBRA algorithm identifies a threshold β—a lower count limit for an independent filtering step above which it is assumed that the read counts follow a negative binomial distribution. This threshold is used for removing results for barcodes with read counts not following negative binomial model and hence possibly incorrectly classified as differentially represented. To find a suitable β for a given data, the DEBRA algorithm samples read count data using a window of N barcodes ordered by their mean count values (Appendix Fig S11). For each sampling step, the algorithm estimates the parameters of the negative binomial (NB) distribution—dispersion (a) and mean (m). DEBRA uses these parameters to generate NB random variables X~NB(m,a) of the same size as the sampled data to calculate theoretical (expected) and empirical two-sample Kolmogorov–Smirnov (KS) test statistics for each sampling window. The KS empirical test statistic was calculated between the sampled values and X~NB(m,a) random variables, while the theoretical KS statistics is calculated between two X~NB(m,a) random variables (see Appendix Fig S12A for examples). The β threshold was estimated by searching for the value of the mean read count at which the overlapping area between the empirical and theoretical density functions of the KS test statistic is close to the maximum overlap for the given data sample. For the estimation, both the theoretical and empirical test statistics are modelled as a Gamma-distributed random variables (see Appendix Fig S13) for each window of size N (here, 30 KS test statistics values on the mean ordered data). The overlap area was calculated separately for each window and then combined from multiple windows by fitting a sigmoid function of mean read counts (see Appendix Fig S12B for examples of fitting the null samples) with four parameters using *drc::drm()* function with *fct = LL.4()* parameter. If the sigmoid curve is ascending and the minimum overlap value is less than 0.25, then β threshold is estimated as the mean count at which the sigmoid-fitted overlap takes the value of 0.8 of the maximum (see Appendix Table S5 for full β threshold estimation rules).

### Dispersion estimation and inference of differentially represented barcodes

To estimate tagwise dispersions, we created a DESeqDataSet object, where we pass only the treatment columns (aka test columns) and calculate the dispersion using *DESeq2::estimateDispersions()* function using the intercept model (*design = ~ 1*) and *fitType = "local"* parameter.

For trended method, we estimate dispersions from a local dispersion trend function as fitted by DESeq2 (parametrization first proposed in DEXSeq (Anders *et al*, 2012)). For calculation, the local dispersion model was obtained from DESeqDataSet *object* by object@dispersionFunction command and used to calculate the tagwise dispersions by providing mean read counts to the obtained fitting function.

The shrunken dispersion estimates were extracted directly from DESeqDataSet object using *DESeq2::dispersions()* function.

The dispersions for barcodes with counts less than β in the test samples were set to the maximum value of the calculated tagwise dispersions to reduce false positives from the barcodes not following NB model if the β thresholding step (aka modified independent filtering) is not used.

In the next step, the previously obtained dispersions were passed to the DESeqDataSet (DESeq2) or CountDataSet (DESeq) object, containing both control and condition columns with *design* = ~ *condition* formula that are required for inference of DRBs. This object was used to test the barcodes for differential representation with either *nbinomWaldTest()* or *nbinomLRT()* tests for DESeq2 implementation or with nbinomTest() for DESeq. Parameter independentFiltering was set to *"FALSE"* when calling *results()* function of DESeq2.

### Independent filtering and β thresholding

We applied the independent filtering procedure (Bourgon *et al*, 2010; Love *et al*, 2014) as a separate function, which uses DESeq2 or DEseq result table as an input. The filtering algorithm uses the *genefilter::filtered_p* function to find the number of null hypothesis rejections at a user-specified FDR cutoff (default parameter is set to 0.2) for the quantiles of the filter statistics (mean read counts). The search algorithm identifies the quantile threshold value that maximizes the total number of rejections in the quantile range of [β,1], where β is the previously estimated threshold for the given data. For the search, the number of rejections is fit as a function of the quantile threshold using a smoothing spline (R function *smooth.spline*), which enables finding the quantile value that corresponds to the maximum number of rejections. User can also set the β threshold value other than the one estimated by the algorithm (see The β threshold estimation section).

### Barcode classification

A barcode is considered to be differentially represented if the Benjamini and Hochberg procedure-controlled FDR is less than a predefined threshold (here, 0.05, 0.10 and 0.25 were tested). If the count fold change between the test and control groups is less than one, then the barcode is considered to be depleted; otherwise, it is classified as enriched. Ground truth for the barcode representation in the perturbed samples was obtained by sequencing the barcode pools (Pool A and Pool B; see Fig 1B), which were used to produce the perturbed samples (Fig 1A). For the ground truth assignments, a barcode is considered enriched if its read Pool A to Pool B count ratio is more than 10; if the ratio is less than 0.1, then the barcode is considered depleted. False positive is defined as a barcode identified by the algorithm as enriched DRB, but which is non-enriched according to the ground truth. For the "random" FDR level, we treated a barcode as enriched if the log fold change was greater than 0.5. Log fold change threshold for the depleted barcodes was set to minus 0.5.

### Precision-recall curves and pAUC calculation

We used precision-recall curves to enable the proper assessment of the test results in samples with varying number of the DRBs (imbalanced dataset). Precision-recall curves were constructed using the "precrec" R package (Saito & Rehmsmeier, 2017). For calculations, the positive class was defined as barcodes correctly assigned by the algorithm to the group it belongs to (enriched or depleted), while the negative class was defined as wrongly assigned barcodes. We used the unadjusted *P*-values for the class assignment by the algorithms, i.e., ranking the barcodes against the ground truth, with low *P*-values indicating high statistical confidence that the barcode belongs to the positive class (i.e. assigned to either enriched or depleted groups by the algorithm). To calculate the precision-recall metrics for simulated experiments with low proportion of enriched barcodes (5%, 15%), we used only barcodes with positive fold change values to assess the algorithms' performance specifically for the enriched barcodes.

We used partial area under the precision-recall curve (pAUC) to compare the relative performance of the algorithms for detecting DRBs in the perturbed samples. The intervals for pAUC calculations were [0,1] for the precision and [0,X] for the recall, where the upper bound X for the recall interval was determined separately for each set of tested samples as the mean recall value at FDR = 0.25 across all the tested algorithms. Such an unbiased selection of the sample-specific recall interval for pAUC calculation allows for comparing the relative performance of the methods in terms of ranking the most significant DRBs for follow-up experiments, especially for the barcodes with low FDR values, without the need for manual selection of the appropriate recall interval for each tested set of samples. The full PR curves are shown in Fig 3D, and Appendix Figs S5B and S6B.

### t-SNE and UMAP algorithms

t-SNE (van der Maaten & Hinton, 2008) was run using Rtsne::Rtsne R function (Krijthe, 2015) with parameter of perplexity = 25 and iterations = 1,000. We set the *pca = FALSE* to disable initial PCA step. UMAP (McInnes *et al*, 2018) algorithm was run using umap::umap R function (Konopka, 2019) with default parameters.

### OVCAR5 multidimensional phenotypic profiling and carboplatin sensitivity experiments

The same pool of barcoded OVCAR5 cells that was produced for generation of the null and perturbed samples were used for OVCAR5 multidimensional phenotypic profiling experiment. The $5 \times 10^7$ cells were taken for the multidimensional phenotypic profiling (T0 time point) experiments, as outlined in Fig EV5.

For carboplatin phenotyping experiments, the cells were barcoded the same way as described for the generation of the null and perturbed samples, and grown to reach an average representation of ~ 1,000 cells per barcode. Cells were counted and split to control, reference and subsample pools, as depicted in Fig EV4.

### Immunostaining

Cells were trypsinized, washed and resuspended in PBS. Then, the cells were fixed and permeabilized with cold 96% ethanol for 30 min on ice, pelleted in a swinging rotor centrifuge at $1,000 \times g$ for 15 min, rehydrated for 30 min in PBS, washed two times in 10 ml of PBS and blocked in PBS with 0.5% BSA for 1 h at room temperature. The staining was done overnight at 4°C in PBS/BSA. Rabbit anti-Ki67 antibody (ab16667, Abcam) was used at 1.5 μg/ml. Following three washes with PBS with 0.5% BSA, the cells were stained with secondary goat anti-rabbit conjugated with Alexa555 at 1/500 for 30 min at room temperature, washed three times and resuspended in PBS for subsequent sorting.

### FACS

All the sorting experiments were carried out using SONY SH800Z Sorter at Biomedicum Helsinki FACS Core Facility, and the data analysis was performed using Sony Cell Sorter software.

### ALDH activity assay

The cells in the log phase of growth were trypsinized and resuspended in medium; the concentration of cells was adjusted to $2 \times 10^6$/ml. The ALDH activity was measured using Aldefluor assay (StemCell Technologies, catalog number 01700) according to the manufacturer's protocol. Cells from the upper and lower quantiles of the Aldefluor fluorescence intensity range were sorted as ALDH$^{high}$ and ALDH$^{low}$ populations, respectively.

### Efflux assay

Cells were trypsinized and resuspended in medium, and the concentration of cells was adjusted to $2 \times 10^6$/ml. The cells were incubated with CDy1 fluorescent dye diluted 1/1,000 (Active Motif, catalog number 895) for 30 min at 37°C in a water bath in the presence or absence of the ABC pump inhibitors tariquidar (1 μM) and probenecid (50 μM). Then, the cells were washed three times in ice-cold PBS and resuspended in medium with or without the drugs. Control cells in medium with efflux inhibitors were left on ice for 2 h, while the test samples were incubated at 37°C for 2 h to allow the efflux of the dye. After three washes, the cells were resuspended in PBS and sorted by the fluorescence intensity in the FL3 (PE-Texas Red) channel. The gating of the efflux-positive cells was set based on the fluorescence intensity of the efflux-inhibited control.

### Autophagy assay

The autophagy was analysed by the ratiometric FACS measurement of the amount of Acridine Orange-stained autolysosomes as described previously (Thomé et al, 2016). Overnight-starved cells were used as a control for the induction of the autolysosomes formation. $4 \times 10^5$ cells with high autolysosomes load and $10^6$ cells with low autolysosomes load were sorted by FACS for subsequent gDNA extraction.

### Proliferation assays

For quantification of the clone proliferation rate, the barcoded OVCAR5 cells were propagated in RPMI-1640 medium for seven passages. Samples for barcode representation analysis were collected at days 0, 5, 8, 11, 14, 18, 25, 29. For the analysis of proliferation rate in validation experiments (Fig 4D), the cells were plated at $2 \times 10^4$ per well in 12-well plates (Costar) and imaged every 4 h in an Incu-Cyte HD live cell analysis system (Sartorius) until the cell confluence of all wells reached 100%. The confluence values during the logarithmic growth phase were used to estimate the population doubling time using the formula $H/Log_2(C_F/C_I)$, where H is elapsed time in hours, $C_F$ is final confluence, and $C_I$ is initial confluence.

### Attachment assay

OVCAR5 cells in the log phase of growth were starved for 16 h in serum-free RPMI supplemented with 2 mM L-glutamine. Upon starvation, the cells were trypsinized, washed in serum-free medium and counted using a Countess II device (Invitrogen). Five millions of live cells were plated in serum-free medium to 15-cm cell culture dishes and allowed to attach for 12 h. Upon incubation, the non-adherent cells were collected for genomic DNA extraction by centrifugation at $500 \times g$ for 5 min. For the validation experiments, non-adherent cells were collected and replated, and both adherent and non-adherent cells were allowed to recover in serum-supplemented medium for 24 h prior to the evaluation of their proliferation rate.

## Data availability

The datasets and computer codes produced in this study are available as following:

- Barcode read counts for benchmark datasets are provided as Datasets EV1–EV3.
- Barcode read counts for OVCAR5 multidimensional phenotyping experiments are provided as Dataset EV4.
- Computer code used to generate the main figures is provided as Code EV1.
- The implementation of the DEBRA algorithm is accessible through Github portal (https://github.com/YevhenAkimov/DEBRA).

**Expanded View** for this article is available online.

## Acknowledgements

The authors thank Pekka Ellonen, Tiina Hannunen and Lars Paulin for the great NGS service. This project has received funding from the European Union's Horizon 2020 research and innovation programme under grant agreement No 667403 for HERCULES. The work was also supported by grants from the Cancer Society of Finland (TA and KW), the Sigrid Jusélius Foundation (TA), Academy of Finland (grants 292611, 310507, 313267, 326238 to TA) and Novo Nordisk Foundation (Novo Nordisk Foundation Center for Stem Cell Biology, DanStem; grant NNF17CC0027852 to KW).

## Author contributions

TA and YA initiated and conceptualized the study; TA and KW supervised the study; YA, TA, DB and KW wrote the manuscript; TA provided critical support in statistical analysis; YA designed the experiments, developed the DEBRA methods, performed statistical analysis and most of the experiments including plasmid generation, library construction, clone-tracing experiments and NGS library preparation with the help of DB and ST; DB designed and performed phenotypic assays for multidimensional phenotypic profiling. All authors discussed the results and approved the final manuscript.

## Conflict of interest

The authors declare that they have no conflict of interest.

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
