## [Review Process File · Molecular Systems Biology]

Improved detection of differentially represented DNA barcodes for high-throughput clonal phenomics

Yevhen Akimov, Daria Bulanova, Sanna Timonen, Krister Wennerberg, and Tero Aittokallio

Review timeline:

Submission date:	22 nd August 2019
Editorial Decision:	1 st October 2019
Revision received:	30 th December 2019
Editorial Decision:	7 th February 2020
Revision received:	13 th February 2020
Accepted:	17 th February 2020

Editor: Maria Polychronidou

Transaction Report:

1st Editorial Decision

1st October 2019

Thank you again for submitting your work to Molecular Systems Biology. We have now heard back from the three referees who agreed to evaluate your study. As you will see below, the reviewers acknowledge that the study seems relevant for researchers performing barcoding experiments and lineage phenotyping. They raise however a series of concerns, which we would ask you to address in a major revision.

I think that the reviewers' recommendations are rather clear and I therefore see no need to repeat the points listed below. Please feel free to contact me in case you would like to discuss in further detail any of the issues raised by the reviewers.

REFeree REPORTS

Reviewer #1:

Akimov et al highlight and address an important but often overlooked facet of DNA barcoding, namely the accuracy of statistical methods used to infer differentially represented barcodes (DRBs). While robust statistical analysis pipelines have been developed for analysing RNA-seq, these methods have not been benchmarked for performance in the context of analysing DNA barcoding experiments. The authors devise a clever strategy to test various analysis pipelines on manually mixed populations of barcoded cells comprising known frequencies of a set of DNA barcodes which they call "null subsamples". Expectedly, the authors find that the variance in read counts for these null samples increases when smaller numbers of cells are isolated for sequencing due to sampling bottlenecks. Furthermore, they determine that at low cell numbers, barcodes with low read counts fail to obey a negative binomial (NB) distribution and thus violate an assumption of existing RNA-

seq analysis pipelines. Consequently, one important finding from this work is a demonstration of the surprisingly high false discovery rate when DESeq, DESeq2 or edgeR are used to quantify DRBs in small cell samples. To develop an analysis process that is less prone to false positives, the authors make two modifications to DESeq. First, they determine dispersion estimates for barcodes based on variation for the experimental condition only rather than considering variation in the control group. Next, they determine a median read-count threshold (which the author's term Beta) above which read counts accurately fit a negative binomial distribution. This new algorithm termed DEBRA appears to produce far fewer false positive DRBs. The authors go on to apply this algorithm to assess phenotypic differences across lineages of an immortalised cell line.

An important conclusion of this study is that false positives are prevalent when RNA-seq algorithms are applied to analyse DNA barcoding experiments with low cell numbers and limited representation of individual barcodes. The authors provide strong support for this conclusion through the use of defined control mixtures of barcoded cells. This limitation was not appropriately appreciated in previous work. The authors also provide convincing support that their new algorithm DEBRA provides more conservative estimates of the number of DRBs. Reporting complete, rather than partial, AUCs will clarify whether this pipeline is more efficient for identifying DRBs overall. Finally, the authors demonstrate that the OVCAR5 cells used in their experiments appear to comprise multiple, phenotypically distinct lineages. This point is of interest as cell lines are often assumed to be phenotypically homogenous. This novel and important study is of interest to researchers performing DNA barcoding experiments. As pooled CRISPR screens are becoming increasingly popular this work may also have applications to assessing sgRNA enrichment/depletion.

Major concerns

1. The partial AUC values presented in figure 3C, S5A and S6A give an overly optimistic impression of the sensitivity of the DEBRA method. Reporting the AUC for the entire recall interval $[0,1]$ would be more appropriate.
2. In the lineage phenotyping experiments the only comparison between DEBRA and previous algorithms is made in Figure S8a and there does not appear to be much difference. How would the results presented in figures 4 and 5 compare if DEseq were used rather than DEBRA? If the lineage phenotyping study is a demonstration of the power of DEBRA over other methods, this comparison is critical. If, on the other hand this is an unrelated exploration into the power of barcoding technology, then more work could help uncover the basis of phenotypic heterogeneity within the cell population analysed. Because the barcoded lineages appear to retain their phenotypic traits over several cell divisions, it should be possible to isolate and expand individual cells from distinct lineages for further clonal analysis to elucidate the genetic, transcriptional or epigenetic differences giving rise to the distinct phenotypic clusters reported in figure 5.

Minor concerns

1. The authors demonstrate that barcodes with low read counts are more likely to differ from an NB distribution. They imply that this is a source of false positives in existing sequencing algorithms. Rather than only showing goodness of fit vs. read count (Fig 2C), the authors should go one step further and explicitly plot the frequency of DRBs vs mean counts for their null subsets. This would provide stronger evidence that false positives indeed arise from barcodes with low read counts that violate the NB assumption.
2. The study explores the impact of sample size and barcode frequency on algorithm performance but have not measured the effect of sequencing depth. This is an important parameter that influences identification of differentially represented genes and barcodes (Li et al. 2014). This paper should at least comment on the sequencing depth and distribution in the study and how this compares to typical barcoding experiments.
3. While the authors provide a nice comparison to commonly used RNA-seq algorithms, these tools were not purpose built for barcoding experiments. It could be interesting to compare DEBRA to more suited algorithms such as those developed for analysing pooled CRISPR screening experiments like MAGeCK (Li et al. 2014). MAGeCK also assumes that barcode frequencies are

NB distributed so this algorithm might also be prone to false positives for barcodes with low read counts at small cell numbers. Application of a Beta threshold, such as the method implemented in DEBRA, could reduce false discovery for this method as well. This analysis is not necessary but could widen the appeal of this paper to the rapidly growing CRISPR screening community.

Reviewer #2:

Here, Akimov et al., present a statistical approach to identify the differential response of DNA-barcoded cell subpopulations to cellular stimuli. The authors use a mix of DNA-barcoded cell pools, adapting existing algorithms developed for RNA-Seq to accurately detect differentially responding lineages. They go on to apply this to assess clonal dynamics in cancer cell lines subjected to a variety of perturbations, demonstrating the utility of this approach for high-throughput lineage phenotyping. We have the following major comments:

- Overall, critical aspects of the study are poorly explained. In the light of alternative single-cell based barcoding approaches, the methods used by the authors need to be presented much more clearly to avoid confusion. This could be an elegant approach but at present it is difficult to follow the methodologies used.
- Given the recent explosion of single-cell technologies, it would be helpful for the authors to expand on their brief mention of single-cell applications in the discussion. For example, what is the advantage of this approach compared to single-cell RNA-based methods of clonal tracking? An upfront discussion of this could better frame the strengths of their approach (which I suspect is throughput, cost-savings etc). In a similar vein, the authors could discuss how effective their statistical approach would be in lineage phenotyping experiments that use RNA barcodes (eg. MPRA's etc.)
- Throughout the manuscript, key information on sample sizes and replicates is not clearly detailed - something I would expect for a statistical approach.
- The authors claim that DEBRA detects differentially responding lineages with higher accuracy. However, a thorough comparison of analysis done using the authors' and existing methods of an actual, hypothesis driven experimental dataset is missing. While the authors compare false positive rates for their and other methods using synthetic and published datasets, it would be more insightful to see the advantages that the reduced false positive rate of their method actually yields, in actual biological inference.
- Again, the authors report that DEBRA works significantly better with trended dispersions as compared to shrunken dispersions, especially for small cell populations. The authors should at least briefly discuss why that is, so that better insight can be gained into their method.
- The text for "Multidimensional phenotypic profiling of barcoded cells identify distinct cancer cell subpopulations" could be better framed. The authors could state the hypotheses behind the series of perturbations performed on OVCAR5 cells upfront and strengthen them with literature citations. This would help define the rationale behind their experiments better, instead of directly diving into the experimental steps for the perturbations performed.

Reviewer #3:

In the above paper the authors devise a benchmark experiment by which to test classic RNA-seq differential expression detection algorithms for their use in barcoded lineage-tracing data. They find that some of the assumptions inherent to these algorithms do not hold in the case of lineage tracing experiments. Specifically, the authors propose a number of modifications of the DEseq algorithm,

collectively termed DEBRA, which improve sensitivity and specificity in precisely these cases. The proposed modifications take the form of a threshold on barcode counts, such that these better follow the assumptions made by DEseq; as well as modifying DEseq's dispersion estimate procedure. In order to demonstrate the improved performance of the proposed method, the authors create two datasets: one each with and without DRBs, and test the performance of different RNA-seq differential detection algorithms in these cases, given the known ground truth from the design of the experiment.

The authors couple this result with an introduction to high-throughput lineage phenomics, wherein single lineages may be clustered by phenotypes measured in different assays, and which opens the possibility of gleaming additional insights about the characteristics of differentially responding lineages in different experiments w.r.t their phenotype. The methods presented here will prove useful to scientists carrying out barcoding experiments and for those who are interested in gaining insights into the phenotypic differences of clones responding differentially to given experimental conditions.

This work follows an overall high standard of experimental planning and computational analysis, yet falls short at several points. Many important details of the experiments and analysis methods are not expounded on in sufficient depth. If the authors were to rephrase parts of the manuscript into a more pointed form, better laying out the background and context of the decisions leading to the results presented in the paper, these would surely benefit future readers in their ability to see the role of these results and apply them in the context of their work. Hence, this reviewer suggests the following major points which should be addressed in the revised manuscript:

Add information about "gold standard" lineage tracing experiment, for example barcode-counts, barcode frequencies by cell-count in subsamples, read-count enrichment ratios in perturbed subsamples, etc. The manuscript lacks details of this purported "gold standard" experiment which hampers the interpretation of key findings, for example the tagwise variation in Fig. 2. Similarly, one might offer additional information regarding perturbed subsample composition, particularly as it differs from the null subsamples.

Include formulae or text describing how DEseq was applied, e.g. $\text{counts}(\text{DRB}) \sim \text{population} + \text{perturbation}$, as well as a fit of the empirical Kolmogorov-Smirnov test as shown on DEBRA's github page in Fig. 2. This would strengthen claims made in association with the failure of barcode counts to conform to a negative binomial distribution.

Adding additional biological replicates of null- and perturbed-subsamples would strengthen both the author's claims about the relationship between cell-count and variance, but also improve DEseq's ability to estimate dispersions and generally reduce the false positive rate.

A "gold standard" lineage tracing experiment is described, but not related to any typical assays carried out in a lab. It would be relevant to know the envisioned scope of DEBRA's application; ideally compounded by the author's insights into the benefits of using their software in real-life scenarios, for example in small molecule drug screens.

Similarly, potential applications of the methods presented here could be made more clear by further expounding upon the benefits, for example to the data used from Seth et al, derived from increased sensitivity in detecting DRBs. Alternatively, it might be prudent to take a closer look at the Seth et al DRB differential and offer some interpretation to the extent that the detected - or false positive - DRBs are biologically meaningful or make sense in the context of the experiment.

Significantly, there appears to be a distinct divide between the methods discussed for DRB detection and their use in high-throughput lineage phenomics. The authors should make explicit the connection between these items, as currently their interaction is left to interpretation by the reader. Indeed, their disjunct placement in the manuscript effectively dilutes the significance of the results of either. This could perhaps be overcome by highlighting the difference in biological conclusions drawn from DRB clusters identified in lineage phenomics applications based on DEBRA and unmodified DEseq variants?

When describing the "trended" vs. "shrunk" DEseq modifications the authors reference the methods section, yet these methods are not discussed there in any further detail. There exists a passing reference in the manuscript's discussion, but which does not contain enough detail to satisfy this reviewer's curiosity concerning the novelty or indeed methodological basis of the suggested modifications to DEseq. The authors could include mathematical formulae to explain these functions conceptually, and having done so, might furthermore address the reduced performance of the "shrunk" version of the algorithm as compared to the "trended" variant, i.e as depicted in Fig. 3a. The implications of forcing the data to conform to the assumptions inherent in DEseq by filtering

non-NB reads could be further expounded upon. The authors claim that "barcode count data [...] seriously violate basic assumptions [of DEseq]"; it is unclear whether forcing conformity in the case of non-NB reads ameliorates the stated violation.

Minor comments

The use of the term 'lineage' more generally refers to a differentiation trajectory of cells; the term 'clones' or 'clonal expansion' might describe the given conditions more precisely.

The distinction from the experimental design proposed here to single cell sequencing experiments can be made explicit, particularly w.r.t the t-SNE in Fig 5a.

Other tested DEseq variants; why does edgeR fail like DEseq does?

Fig. 3c: A classical ROC-curve plot annotated with the AUC would provide a better overview of the differences in performance of the individual methods and highlight the benefit provided in DEBRA

The role of Fig 3d and e are unclear in the context. There is no ground truth known and no interpretation as to the observed differences in DRB detection offered, hence these figures might better fit in the supplement.

It is unclear how the validation of the phenotypic profiling (Fig. 4) relates to the improvement in DRB detection. If there is no tangible connection, the manuscript's cohesion might be improved by combining Figures 4 and 5.

The clusters identified in Fig 5a are not clearly defined. Perhaps the authors can repeat the clustering with other projections such as PCA, UMAP, or MDS, either to identify novel clusterings or demonstrate the stability of existing clusters.

Additionally describe generation of read count tables, what barcode ratios represent, and why DEBRA will be better able to approximate them .

Was any preprocessing/PCA etc of phenotype vectors carried out before running t-SNE?

Lineage read count cutoff of 70 in Fig. 4 vs 75 in Fig. 5.

the interpretation of this sentence is unclear: "10⁵ times less of [sic] individual sequencing tags as compared to an RNA-seq sample produced from the same number of cells".

The Discussion mentions that simulated effect sizes were low, yet it is implied that the experimental setups that the simulated lineage tracing experiment were intended to follow are carried out in the context of "very narrow sample size bottlenecks", i.e high effect sizes. Are higher perturbation degrees in the simulated experiment required in order to accurately reflect real experimental conditions?

In conclusion, the proposed method provides the means to better detect DRBs in lineage tracing experiments and enables querying the phenotypic landscape of individual lineages in a high-throughput phenomics approach. The manuscript lacks clarity and would benefit from the addition of information regarding the experimental setup and proposed method, in particular as DRB detection relates to the described lineage phenomics analysis.

1st Revision - authors' response

30th December 2019

Reviewer #1:

1. The partial AUC values presented in figure 3C, S5A and S6A give an overly optimistic impression of the sensitivity of the DEBRA method. Reporting the AUC for the entire recall interval [0,1] would be more appropriate.

Our response: Thank you for the suggestion. We used partial AUC values since these are widely used to evaluate overall precision of methods, when the focus is on the findings (here, DRBs) that are ranked at top by the method, and that often are being then subjected for follow-up experiments (<https://doi.org/10.1038/s41467-019-12266-7>; <https://doi.org/10.1038/s42003-019-0415-5>; <https://doi.org/10.1093/nar/gkw521>). This is the practical application of the DRB detection methods, i.e., to prioritize the top DRBs with the highest precision, and therefore we feel pAUC provides a reasonable scoring metric for

comparing the performance of the DRB detection algorithms. We have also provided the **full PR curves** of the methods comparisons (Fig. 3D; Appendix Fig S5B and S6B) to show the performance of the algorithms across the full recall interval [0,1].

However, we also agree with the comment that the fixed recall interval of [0,0.25], that was used in the original figures, may give an overly optimistic impression of the sensitivity of the DEBRA method. In response to this point, we first describe below the specific calculations and reasons why we have chosen to use the partial AUC as the primary evaluation score in our comparison, and then we describe the modifications in the calculations we did in the revised manuscript to improve the comparison results and their interpretation.

We chose to use the area under a precision recall curve (AUC) as a performance metric to investigate whether the DEBRA-estimated p-values are better at prioritizing DRBs as compared to those estimated by the original RNA-seq analysis algorithms. The full AUC is calculated by ranking all the DRBs according to their p-values, which takes into account all the barcodes regardless of their significance level (here, FDR value). Hence, the full AUC also considers the rank ordering of barcodes with high FDR values that are not very informative for the practical application or when interpreting the results. This drawback of the full AUC has been described, for instance, by Jorge Lobo et. al (2008) [<https://doi.org/10.1111/j.1466-8238.2007.00358.x>].

Please see Appendix Fig S5B (e.g., samples with perturbation 18, size 20; perturbation 18, size 40; or perturbation 27, size 20) as examples of how full AUC may lead to a non-practical performance comparison that would lead the reader to make a wrong conclusion. In these examples, the modified DESeq2 algorithms show better performance than the original DESeq at low recall values (and hence, low FDR values). However, the original DESeq starts to score barcodes better than the modified DESeq2 in the high FDR region (that are usually not of interest for a researcher). Therefore, calculating AUC for the entire recall interval [0,1] would score original DESeq performance similar to DEBRA, even though it performs much worse in the low FDR region (the region the researcher is interested in).

To address this comment, we have made the following changes in the revised manuscript:

- a. Instead of using a fixed recall interval for all samples, we have now estimated an appropriate sample-specific recall interval for the pAUC calculations. Specifically, for each sample, we calculate pAUC based on [0,X] recall interval, where X is the mean recall value at FDR=0.25 across all the methods tested in a given sample. We believe this approach provides a fair and unbiased scoring metric, as it prevents “oversaturating” the metrics (see figure below) observed in previous version of the figures. **As a result, we have modified Fig 3C, Appendix Fig S5, and Appendix Fig S6.**

- b. To prevent the reader from mistakenly overestimating DEBRA performance, we have **included** the corresponding full **precision-recall curves to Fig 3D**.
- c. We have better described the reasons for using pAUC metric in the **Methods** section, and detailed the calculation of the sample-specific pAUC scores (page. 20, par 4).

Effect of the recall interval estimation strategies on the pAUC calculation.

Comparison of pAUC values estimated based on a fixed recall interval of $[0, 0.25]$ (left) vs sample-specific recall-interval of $[0, X]$ (right; this is now used for all the pAUC calculations in the revised manuscript), where X is estimated for each sample separately as the mean recall value at $FDR=0.25$ across all tested algorithms.

(A) 0.5 proportion of enriched barcodes, or

(B) 0.15 proportion of enriched barcodes

2. In the lineage phenotyping experiments the only comparison between DEBRA and previous algorithms is made in Figure S8a and there does not appear to be much difference. How would the results presented in Figures 4 and 5 compare if DESeq were used rather than DEBRA? If the lineage phenotyping study is a demonstration of the power of DEBRA over other methods, this comparison is critical. If, on the other hand, this is an unrelated exploration into the power of barcoding technology, then more work could help uncover the basis of phenotypic heterogeneity within the cell population analysed. Because the barcoded lineages appear to retain their phenotypic traits over several cell divisions, it should be possible to isolate and expand individual cells from distinct lineages for further clonal analysis to elucidate the genetic, transcriptional or epigenetic differences giving rise to the distinct phenotypic clusters reported in figure 5.

Our response: We agree that in the original version of the manuscript, the link between the DEBRA method development part and the “multidimensional phenotypic profiling” section was unfortunately not clear enough. To improve on this issue, we have made the following modifications to the manuscript:

1. We have performed additional experiments in which we demonstrate that DEBRA improves the DRB detection, compared with the original algorithms, also in the context of real selection pressure (carboplatin treatment). **The new result section can be found on pages 8 (and in new Fig 4, Fig EV4 and Appendix Fig 9).**
2. We compared the clustering of clones selected by p-values produced by the DEBRA and DESeq2 algorithms (**Fig EV6C**). Clones selected on the basis of the DEBRA-derived p-values showed more robust clusters on the UMAP projection, highlighting the importance of proper statistical inference in such downstream analyses. **This result is briefly mentioned in the revised results section (page 10, par. 3).**

We would also like to note that a proper significance testing enabled by DEBRA is essential for a researcher to guarantee that the observed clonal responses are at least statistically significant, and hence, can be used for multidimensional phenotypic profiling or other downstream applications. In this sense, the DEBRA analysis is a critical prerequisite for the downstream analyses such as multidimensional phenotypic profiling and clone-specific follow-up studies (e.g. subcloning and subsequent molecular profiling). **This is now briefly mentioned in the revised manuscript (page 12, par. 3) and in several places throughout the manuscript .**

We agree with the Reviewer that subcloning might provide a way to validate the phenotype associations observed in the multidimensional phenotyping approach. However, in the light of recent findings [<https://doi.org/10.1242/jcs.217661>], suggesting

that clones grown in isolation from the population (subcloned) may substantially differ from those grown in the population, we believe that validation via subcloning may not become conclusive. Instead, we feel the proposed approach of multidimensional phenotypic profiling was sufficiently validated by the independent measurements of the phenotype-phenotype associations (Fig 5D,E), which recapitulated the known KI67 - proliferation relationship (Fig 5C), and by the identified clusters which recapitulated the well-known phenotype associations (e.g. autophagy/ALDH; drug resistance/ALDH; Efflux/drug resistance; ALDH/population size). We think that showing such a novel application of the DRB detection algorithm in a practical biological application is needed in this kind of a methodology paper, in order to demonstrate how the algorithm enables new biological discoveries and future high-resolution applications.

Minor concerns

1. The authors demonstrate that barcodes with low read counts are more likely to differ from an NB distribution. They imply that this is a source of false positives in existing sequencing algorithms. Rather than only showing goodness of fit vs. read count (Fig 2C), the authors should go one step further and explicitly plot the frequency of DRBs vs mean counts for their null subsets. This would provide stronger evidence that false positives indeed arise from barcodes with low read counts that violate the NB assumption.

Our response: We thank the Reviewer for this valuable comment that pointed out we did not make this point clear enough in the original version. We tried to avoid of making the claim that the main source of false positives from original RNA-seq algorithms originate from barcodes which do not follow the NB distribution. Rather, we show that there are two main discrepancies between the data assumptions made by RNA seq analysis algorithms and the real barcode sequencing data - (i) variance differences between control and test samples (Fig 2AB), and (ii) read counts deviance from NB model in a low count region (Fig 2C). Our main hypothesis was that false positives primarily arise due to former (i), which leads to incorrect estimation of the tagwise dispersion (because of large sample-specific differences in variance, please see Fig. 1A).

However, as highlighted by the Reviewers' comment, it was not very clearly described in the text of the originally-submitted manuscript. In the current version, **we clarified this (page 5-6)** to make it clear for the readers. **We have also included new Appendix Fig S4 (also attached below)**, where we show that proper estimation of the dispersion (addressing discrepancy i) is enough to eliminate most of the false discoveries. The remaining false discoveries are eliminated when the β thresholding is applied. In the new set of experiments made for the revised manuscript (**new Fig 4, Fig EV4, Appendix Fig 9 and new results section on page 8-9**), we observed that barcodes with read counts lower than the β threshold may significantly contribute to the false discovery rate via unexpected performance of the dispersion fitting algorithms (see Appendix Fig S9; also attached below).

Appendix Fig S4

The effect of the modifications implemented in DEBRA on the false discovery rate (FDR).

MA-plots for the DRB detection results with various versions of the algorithms (top) and their DEBRA modifications (right) on the null-20 subsample (two replicas) tested against four null subsamples (two null-660 and two null-330). Barcodes with $\text{FDR} > 0.1$ are marked in red.

Appendix Fig S9. Characteristics of the subsample-specific DRBs in the carboplatin response phenotyping experiment

- Distribution of the mean counts of the DRBs detected in carboplatin response sensitivity experiment with original DESeq2(Wald) or DESeq in the subsamples (reduced sample size) but not in the reference samples (indicating that most of them are likely to be false discoveries). The dashed red line indicates the β threshold.
- Mean - dispersion relationship as fitted with original DESeq or DESeq2 for subsample size of 1.3 million cells. Significantly lower dispersion trend for DESeq in a low read count region is likely a cause of the observed discrepancies between original DESeq and DESeq2 algorithm. Dispersion fits were extracted using `object@dispersionFunction` in DESeq2 and using `object@fitInfo[["pooled"]][["dispFunc"]]` in DESeq.

2. The study explores the impact of sample size and barcode frequency on algorithm performance but have not measured the effect of sequencing depth.

This is an important parameter that influences identification of differentially represented genes and barcodes (Li et al. 2014). This paper should at least comment on the sequencing depth and distribution in the study and how this compares to typical barcoding experiments.

Our response. Thank you for the suggestion. We have now **included new Appendix Figure S4**, where we show the dependence of DEBRA performance on the sequencing depth. This new analysis shows that DEBRA is able to faithfully control for false discoveries under different sequencing depths (**mentioned on page 7, second paragraph**).

3. While the authors provide a nice comparison to commonly used RNA-seq algorithms, these tools were not purpose built for barcoding experiments. It could be interesting to compare DEBRA to more suited algorithms such as those developed for analysing pooled CRISPR screening experiments like MAGeCK (Li et al. 2014). MAGeCK also assumes that barcode frequencies are NB distributed so this algorithm might also be prone to false positives for barcodes with low read counts at small cell numbers. Application of a β threshold, such as the method implemented in DEBRA, could reduce false discovery for this method as well. This analysis is not necessary but could widen the appeal of this paper to the rapidly growing CRISPR screening community.

Our response: It is true that the RNA-seq algorithms were not initially developed for barcoding experiments; nevertheless, they are being used in the detection of DRBs, and this was the motivation to show their sub-optimality and improve their performance by DEBRA.

The MAGeCK algorithm for CRISPR screen implements a two-step significance testing for assessing the effect of gene KO/activation on the cells' phenotype in CRISPR/Cas9 screening experiments.

First, MAGeCK estimates sgRNA-specific p-values in a manner similar to DESeq (and other conditioned tests). MAGeCK have shown to provide a slight improvement in FDR control compared to the original DESeq algorithm (see Li et. al. 2014; Table M1), but the paper does not elaborate on what modifications led to this improvement and how it is related to the properties of the barcode count data. Most probably, the improvement is due to the better mean-variance modelling which can be seen in MAGeCK original paper (Li et. al. 2014; Supplementary Figure 1).

The second main step in MAGeCK workflow is aggregation of sgRNA p-values to produce gene-wise p-values using an order statistics (i.e. aRRa), which uses a robust rank aggregation algorithm to combine the rank information of sgRNAs targeting the same gene, and to produce finally gene-specific p-values. To our understanding, this is what makes MAGeCK useful for the CRISPR screen analysis; however, this step is not applicable to the clone-tracing experiments as there is only one sequencing tag for each clonal lineage.

Therefore, we did not include MAGeCK into our comparison.

To demonstrate that the MAGeCK does not address the issues raised in our manuscript (that were addressed by DEBRA algorithm), we compared two Null-20 (test) samples with two Null-660 (control) using MAGeCK algorithm. In the sgRNA summary (barcode sequences were used as a sgRNA IDs) the proportion of DRBs was as follows:

34% (FDR<0.25)

23% (FDR<0.10)

20% (FDR<0.05).

Hence, MAGeCK demonstrate similar performance to the DESeq2 algorithm by detecting high proportion of false DRBs in the null samples (no barcode is expected to be differentially represented in the null samples)

We have now cited the MAGeCK and BAGEL algorithms for CRISPR screen analysis in the discussion (**page 13, par. 1**), where we proposed that DEBRA can be used for the analysis of positive selection CRISPR screens.

Reviewer #2

- Overall, critical aspects of the study are poorly explained. In the light of alternative single-cell based barcoding approaches, the methods used by the authors need to be presented much more clearly to avoid confusion. This could be an elegant approach but at present it is difficult to follow the methodologies used.

Our response: We thank the Reviewer for appreciating the importance of our approach, and do apologize for its sub-optimal description in the originally-submitted version. **We have now modified Fig 1, Methods section (page 17, par. 2; page 18, par. 3) and throughout the introduction (pages 2-3) to clarify our approach.** We do hope that these changes have made the presentation clearer.

We would like to also note that DEBRA algorithm was developed for the well-established lentivirus-based cellular DNA barcoding approach. Since first introduced in 2010, it has become a widely adopted approach in many biomedical applications. We have provided representative examples of studies employing this approach in the first paragraph of the introduction (p. 2). These studies use lentiviral integration of unique DNA sequences for labeling the cells, followed by selection for the phenotype of interest. Then DNA amplicon sequencing (via PCR barcode amplification from genomic DNA) is used to estimate the change in clone sizes, and hence to quantify clone-specific differences in the phenotype of interest. In our study, we have developed an extended cellular DNA-barcoding approach towards a general experimental and analytical framework, and implemented an algorithm (DEBRA) for testing the significance of the clonal phenotypes.

- Given the recent explosion of single-cell technologies, it would be helpful for the authors to expand on their brief mention of single-cell applications in the discussion.

Our response: Thank you for the suggestion. We have now expanded the discussion on integration of multidimensional single-lineage phenotyping with single-cell applications (p. 13).

For example, what is the advantage of this approach compared to single-cell RNA-based methods of clonal tracking? An upfront discussion of this could better frame the strengths of their approach (which I suspect is throughput, cost-savings etc).

Our response: We would like to elaborate on the differences between scRNAseq-based lineage trajectory tracking, and the clone tracing experiments via cellular DNA barcoding, which was the topic of this work. The scRNA sequencing provides a snapshot of the RNA expression heterogeneity. It has been used to reconstruct cellular lineage trees (e.g. “RNA velocity”, <https://doi.org/10.1038/s41586-018-0414-6>). These trees show the inferred routes by which one cell type is being converted (mature) to another cell type or, in other words, the presumed flow of the transcriptional changes in pseudotime. This approach, unlike cellular DNA barcoding, cannot trace history of a particular clone over time or measure the change in the clone sizes in response to a perturbation. For these reasons, we believe that these approaches are not directly comparable.

However, we do agree that pointing out the related scRNA-seq-based methods is important for those readers who are not experts in the field, and therefore have **emphasized the difference between the two methods in the introduction (page 2, par. 3)**.

In a similar vein, the authors could discuss how effective their statistical approach would be in lineage phenotyping experiments that use RNA barcodes (eg. MPRA etc.)

Our response: We could not unfortunately find any studies employing MPRA for lineage phenotyping. If the Reviewer can provide some references, we would be happy to take a read and see how DEBRA could be applied to such experiments.

To our knowledge, the use of RNA barcodes (similar to MPRA) in combination with clone tracing is only a recently emerging strategy for tracing clonal gene expression over time or in response to stimuli (<https://doi.org/10.1186/s13059-019-1699-y>; <https://doi.org/10.1038/s41586-018-0744-4>; <https://doi.org/10.1101/467886> ; <https://doi.org/10.1101/761981>). The experiments reported in these studies mostly focus on the analysis of scRNA gene expression data, so, appropriate scRNA-seq gene expression analysis algorithms seems to be more suitable for these experiments. However, this approach (readout of the expressed RNA barcodes via scRNAseq) can be combined with multidimensional phenotypic profiling, as proposed in our manuscript. Such combination, in our opinion, should hold a great promise for determination of the

phenotype - expression relationships. This is possible since barcode expressed as RNA molecule can be read out by both DNA amplicon sequencing (classical clone tracing) and scRNA sequencing from the same cell population). In such a combination, the readout via scRNAseq would provide both a single cell expression profiles and clone identities. These clone identities can be then used to link phenotype measured with clone tracing to the expression profile of single cells. Therefore, reliable calling of phenotypically distinct clones (using DEBRA) will be critical in such experiments for ensuring the validity of the identified phenotype - gene expression relationships. Such combination of methods in our view has a great potential for linking phenotypes (e.g. drug response) to the expression profiles of single cells. **We have now expanded the discussion section to include these further aspects (page 13, par. 2).**

We would also like to note that quantification of the expressed barcodes (as RNA molecule) via classical RNA sequencing technology (not scRNA sequencing) for the clone tracing experiments seems sub-optimal. The major concern is that, in such an approach, RNA barcode expression may be subject to additional sources of variation due to locus/cell/lineage-specific differences. In our opinion, amplicon libraries prepared from genomic DNA is preferred over RNA libraries for barcode quantification in clone tracing experiments. To our knowledge, all studies employing cellular DNA-barcoding use genomic DNA as a source for barcode amplification.

- Throughout the manuscript, key information on sample sizes and replicates is not clearly detailed - something I would expect for a statistical approach.

Our response: Thank you for pointing out this deficiency. We have now added the missing information on sample sizes and replicates. The sample sizes and number of replicates are detailed, for instance, in **Fig 1B, Fig EV4, Fig EV5 and in Appendix Table S1, S2 and S3.**

- The authors claim that DEBRA detects differentially responding lineages with higher accuracy. However, a thorough comparison of analysis done using the authors' and existing methods of an actual, hypothesis driven experimental dataset is missing. While the authors compare false positive rates for their and other methods using synthetic and published datasets, it would be more insightful to see the advantages that the reduced false positive rate of their method actually yields, in actual biological inference.

Our response: Thank you for the comment. We would first like to note that the perturbed and null samples are not synthetically generated (i.e. not simulated on computer). The benchmarking datasets used in the original work were all generated from actual clone tracing experiments. To produce the benchmarking dataset, we introduced known clone-specific perturbations by mixing cells from two independent barcoding experiments in defined ratios, which provided us with known ground truth for barcode representations (**see the modified Fig. 1 for details**).

We agree with the Reviewer that testing the DEBRA only in datasets with manually introduced perturbations may be non-sufficient to claim the improved performance of the algorithm for actual phenotyping experiments. For this reason, we undertake an additional experiment which validated the improved performance of the DEBRA in the context of carboplatin drug treatment as a perturbagent. **These new experimental results are shown in new Fig 4, Fig EV4, Appendix Fig S8,9 and described in a new results section (page 8-9)**

- Again, the authors report that DEBRA works significantly better with trended dispersions as compared to shrunken dispersions, especially for small cell populations. The authors should at least briefly discuss why that is, so that better insight can be gained into their method.

Our response: We agree that this result needed more discussion, something we originally ignored due to the technical nature of this result. We have now expanded the part of the discussion section where we discuss a possible cause for the observed differences (**page 12, par. 2**). and expanded this description in our response below.

In the originally-submitted manuscript, we placed a focus on two methods of dispersion estimation - trended and shrunk. Both methods are widely used in the RNA-seq analysis algorithms, and we do not introduce any new modifications to the dispersion estimation workflow itself.

To illustrate the differences between the two methods for dispersion estimation we adopted (see below) the dispersion plot produced by DESeq2 from its on-line the manuals (<http://www.sthda.com/english/wiki/rna-seq-differential-expression-work-flow-using-deseq2>). In this plot, each dot represent a gene/tag. Black dots are the corresponding tagwise dispersions calculated from count values of that *gene only*. The large variance in the “*gene only*” dispersion estimates are due to the low number of points for dispersion estimation (typically only few replicas are used in RNA seq experiments). Then, the information from multiple genes is pooled together to fit a mean-variance trend (red dots). These estimates are referred to as “trended” in our manuscript. The trended method assumes a strict relationship between mean and variance. However, it is assumed that in the RNA seq experiments, the gene variance might deviate from the mean-variance trend due to biological variation between replicas, therefore, it has been proposed to shrink the individual estimates towards mean-variance trend (blue dots), while allowing for outliers (blue circles). The blue dots and circles represent a final tagwise estimates in the DESeq2 workflow, and are referred to as “shrunk” in our manuscript.

Mean - dispersion plot generated with DESeq2.

The question still remains whether or not the deviance from the mean-variance trend, and hence, dispersion outliers can appear in the DNA barcoding experiments. The improved performance of the trended method shown in our work may be attributed to the absence of such an effect in the benchmark dataset; however, we suppose it might happen if the selection is followed by a long re-growth phase. In this case, one would expect shrunken dispersion estimates to perform better than trended. Overall, further studies are needed to elucidate the relative benefits of either method and the functionality of the DEBRA package will be useful for doing so.

- The text for "Multidimensional phenotypic profiling of barcoded cells identify distinct cancer cell subpopulations" could be better framed. The authors could state the hypotheses behind the series of perturbations performed on OVCAR5 cells upfront and strengthen them with literature citations. This would help define the rationale behind their experiments better, instead of directly diving into the experimental steps for the perturbations performed.

Our response: Thank you for the comment, and we completely agree that this section was left somewhat poorly motivated in the originally-submitted manuscript. **We have now substantially revised the text and figures for the multidimensional phenotypic profiling section (p. 9-10)**, as well as added reference citations. We hope these modifications become useful for the overall clarity and structure of the manuscript.

Reviewer #3

Add information about "gold standard" lineage tracing experiment, for example barcode-counts, barcode frequencies by cell-count in subsamples, read-count enrichment ratios in perturbed subsamples, etc. The manuscript lacks details of this purported "gold standard" experiment which hampers the interpretation of key findings, for example the tagwise variation in Fig. 2. Similarly, one might offer additional information regarding perturbed subsample composition, particularly as it differs from the null subsamples.

Our response. We thank the Reviewer for the comment, and agree that these aspects were not optimally described in the originally-submitted manuscript. **We have now added new Fig 1C,D and Fig EV1** to provide additional information on the relative clone sizes (barcode frequencies) characteristics of the benchmark datasets. Please see also **Fig EV4, Fig EV5, Fig 5 and in Appendix Tables S1, S2 and S3**

Include formulae or text describing how DEseq was applied, e.g. $\text{counts}(\text{DRB}) \sim \text{population} + \text{perturbation}$, as well as a fit of the empirical Kolmogorov-Smirnov test as shown on DEBRA's github page in Fig. 2. This would strengthen claims made in association with the failure of barcode counts to conform to a negative binomial distribution.

Our response: As requested, we have now added the design formulae used to call DRBs by DESeq, DESeq2 and edgeR **methods (page 18, par. 4; page 19, par.6)**. We used a simple " \sim condition" ($\text{counts} \sim \text{condition}$) formula to compare representation of barcodes across conditions.

The fits of the Kolmogorov-Smirnov tests for the null subsamples can be found in **Appendix Fig S12**.

Adding additional biological replicates of null- and perturbed-subsamples would strengthen both the author's claims about the relationship between cell-count and variance, but also improve DEseq's ability to estimate dispersions and generally reduce the false positive rate.

Our response: We thank the Reviewer for the suggestion. Indeed, in the original version of the manuscript, we provided only two plots where relationship between cell-count and variance can be observed. One plot for the null subsamples calculated from replica #1 and the other plot for the Seth et. al. PDX data. However, we believe that the current number of samples and biological replicates (6 null subsamples in two replicas and 12 perturbed subsamples in two replicas for OVCAR5 barcoding experiment; 2 null subsamples in 3 replicas for Mia-PaCa-2 barcoding experiment) is enough to demonstrate the existence of the sample size - variance relationship. **To better illustrate this, we have added new Fig EV2A (also included below), where we show the same relationship for other samples and replicas.**

We agree with the reviewer that increasing the number of replicas should lead to the better tagwise variance estimation and, hence, lower number of false discoveries. However, the expected improvement of the original algorithms' performance is unlikely to be detected as it will be completely overwhelmed by the detrimental effect of the improper variance estimation arising from the variance differences between control and test samples. **We have substantially revised the corresponding section (page 5-6)** to clarify the dominating role of improper variance estimation of the false discovery rate. We also note that the current state-of-art procedures for dispersion estimation efficiently estimate variances even for the low number of replicas, thanks to the efficient procedures of the information sharing across the genes/tags.

We kindly ask the reviewer to compare the Fig 2D and Appendix Fig S1A (Design C) to see that false discovery rate does not significantly depend on the number of samples/replicas provided for dispersion estimation, when the non-modified algorithms are used. There were 4 samples used for dispersion estimation in Appendix Fig S1A (Design C), as compared to 6 in Fig 2D, and still there are way less false discoveries in Design C (Appendix Fig S1A). **As we describe in revised manuscript (page 5-6)**, the major factor influencing false discovery rate for original (non-modified) RNA seq algorithms is the difference in the variance between control and treatment samples and, hence, wrong parametrization of the NB model used for significance testing.

Finally, we note that it is important for a DRB detection algorithm to work with low number of replicates due to intrinsic limitation of the material in typical clone tracing experiments (discussed on page 11)

Fig EV2A. Mean-variance relationships for the null subsamples of different sizes

A "gold standard" lineage tracing experiment is described, but not related to any typical assays carried out in a lab. It would be relevant to know the envisioned scope of DEBRA's application; ideally compounded by the author's insights into the benefits of using their software in real-life scenarios, for example in small molecule drug screens. Similarly, potential applications of the methods presented here could be made more clear by further expounding upon the benefits, for

example to the data used from Seth et al, derived from increased sensitivity in detecting DRBs. Alternatively, it might be prudent to take a closer look at the Seth et al DRB differential and offer some interpretation to the extent that the detected - or false positive - DRBs are biologically meaningful or make sense in the context of the experiment.

Our response: We thank Reviewer for this advice, and have accordingly added to the discussion a description of what benefits DEBRA offers for the clone tracing field (**page 12, par. 3**). We have also discussed potential benefits of reliable DRB calling via DEBRA in applications integrating single cell technologies and clone tracing (**p. 13, par. 2**). Overall, the DEBRA is expected to be widely applicable for any clone tracing experiment for the detection of differentially responding clones.

We would like to note that we aimed to design the benchmark experiments to resemble the typical clone tracing experiments carried out in the lab such as those cited in the first paragraph of the introduction. To ensure that the benchmark dataset would have representative barcode distribution, we used cell pools (Pool A and B in Fig 1B) produced from in-vitro clone tracing experiments to recapitulate the real-life clone size distributions.

As suggested by the Reviewer, we have carefully read the Seth et. al. study to see if DEBRA can offer additional value for the interpretation of the presented results. While we appreciate the approach and amount of work the authors performed in this study, they unfortunately performed the validation experiments on a limited number of isolated clones (2 sensitive and 2 resistant) with no proper quantification of the clonogenic assays and missing controls (e.g. Fig S5H misses the controls, response of a bulk is also missing). Hence, we would prefer to refrain from making any further biological conclusions on the possible benefits the DEBRA based on biological conclusions of this study.

Significantly, there appears to be a distinct divide between the methods discussed for DRB detection and their use in high-throughput lineage phenomics. The authors should make explicit the connection between these items, as currently their interaction is left to interpretation by the reader. Indeed, their disjunct placement in the manuscript effectively dilutes the significance of the results of either. This could perhaps be overcome by highlighting the difference in biological conclusions drawn from DRB clusters identified in lineage phenomics applications based on DEBRA and unmodified DEseq variants?

Our response: We agree that in the original version of the manuscript, the link between the DEBRA method development part and the “multidimensional phenotypic profiling” section was unfortunately not clear enough. To improve on this issue, we have made the following modifications to the manuscript:

- a. We have performed additional experiments in which we demonstrate that DEBRA improves the DRB detection, compared to the original algorithms, also in the context of real selection pressure (carboplatin treatment). **The**

new section can be found on pages 8-9 and new Fig 4, Fig EV4 and Appendix Fig 9.

- b. We compared the clustering of clones selected by p-values produced by the DEBRA and DESeq2 algorithms (Fig EV6C). Clones selected on the basis of the DEBRA-derived p-values showed more robust clusters on the UMAP projection, highlighting the importance of proper statistical inference in such downstream analyses. **This result is briefly mentioned in the revised results section (page 10, par. 3).**

When describing the "trended" vs. "shrunk" DEseq modifications the authors reference the methods section, yet these methods are not discussed there in any further detail. There exists a passing **reference** in the manuscript's discussion, but which does not contain enough detail to satisfy this reviewer's curiosity concerning the novelty or indeed methodological basis of the suggested modifications to DEseq. The authors could include mathematical formulae to explain these functions conceptually, and having done so, might furthermore address the reduced performance of the "shrunk" version of the algorithm as compared to the "trended" variant, i.e as depicted in Fig. 3a.

Our response: We thank the reviewer for pointing out this deficiency. We apologize for not clearly describing the modifications in the originally-submitted manuscript. Indeed, in the previous version of the manuscript, it was unclear that the implemented methods for dispersion estimation are not the modifications to the existing methods but, rather, testing of the previously used dispersion estimation methods for the DRB detection by DEBRA-modified algorithms. **We have now substantially revised the description of the modifications implemented into DEBRA algorithm to make it clear (page 5-6). We have also included a short discussion (page 12, par. 2) on "shrunk" and "trended" dispersion estimation**, and expanded this description in our response below.

In the originally-submitted manuscript, we placed a focus on two methods of dispersion estimation - trended and shrunk. Both methods are widely used in the RNA-seq analysis algorithms, and we do not introduce any new modifications to the dispersion estimation workflow itself. The DEBRA introduces two novel modifications to the original DESeq/DESeq2 algorithm. First is that tagwise dispersions for inference of the DRBs is estimated only from the replicas of the treatment sample and, second is the b-thresholding (**please see description on p.5-6 and p.19-20**). Also, in DEBRA, we provided the functionality to independently select from 2 methods of variance estimation ("trended" and "shrunk"), and 3 methods of statistical significance testing (original DESeq method, Wald(DESeq2 implementation) and LRT(DESeq2 implementation)), hence providing a total of $2 \times 3 = 6$ options, something which was not possible with the original algorithms. This should enable further exploration of the relative benefits of these variants in other datasets.

To illustrate the differences between the two methods for dispersion estimation we adopted (see below) the dispersion plot produced by DESeq2 from its on-line the manuals (<http://www.sthda.com/english/wiki/rna-seq-differential-expression-work-flow-using-deseq2>). In this plot, each dot represent a gene/tag. Black dots are the corresponding tagwise dispersions calculated from count values of that *gene only*. The large variance in the “*gene only*” dispersion estimates are due to the low number of points for dispersion estimation (typically only few replicas are used in RNA seq experiments). Then, the information from multiple genes is pooled together to fit a mean-variance trend (red dots). These estimates are referred to as “trended” in our manuscript. The trended method assumes a strict relationship between mean and variance. However, it is assumed that in the RNA seq experiments, the gene variance might deviate from the mean-variance trend due to biological variation between replicas, therefore, it has been proposed to shrink the individual estimates towards mean-variance trend (blue dots), while allowing for outliers (blue circles). The blue dots and circles represent a final tagwise estimates in the DESeq2 workflow, and are referred to as “shrunk” in our manuscript.

Mean - dispersion plot generated with DESeq2.

The question still remains whether or not the deviance from the mean-variance trend, and hence, dispersion outliers can appear in the DNA bacoding experiments. The improved performance of the trended method shown in our work may be attributed to the absence of such an effect in the benchmark dataset; however, we suppose it might happen if the selection is followed by a long re-growth phase. In this case, one would expect shrunk dispersion estimates to perform better than trended. Overall, further studies are needed to elucidate the relative benefits of either method and the functionality of the DEBRA package will be useful for doing so. **These considerations are included in the discussion (page 12, par. 2).**

The implications of forcing the data to conform to the assumptions inherent in DESeq by filtering non-NB reads could be further expounded upon. The authors claim that "barcode count data [...] seriously violate basic assumptions [of DESeq]"; it is unclear whether forcing conformity in the case of non-NB reads ameliorates the stated violation.

Our response: To address this valuable comment, **we have added Appendix Fig S4** to demonstrate the relative effects of the introduced modification on the false discovery rates. **These new results are described on page 6 (last paragraph).** Also, in the new set of experiments described in the revised manuscript (**page 8**) we observe (see Appendix Fig S9) that barcodes with read counts lower than the β threshold may significantly contribute to the false discovery rate and result in an unexpected performance of the dispersion fitting algorithms (addressed by beta-thresholding in DEBRA)

The significance testing by the original RNA-seq algorithms assume that counts for each tag follow the NB distribution, hence removing these tags from the final results (e.g. via β thresholding) will remove possible false discoveries, while not affecting p-values estimated for other tags (since the algorithm assesses the significance independently for each tag). The similar procedure is actually made default in the DESeq2 algorithm. It removes low expressed genes/tags before p-value adjustment (BH) for multiple hypothesis testing to increase the total number of tags for a particular FDR threshold (the procedure is called independent filtering (<https://doi.org/10.1073/pnas.0914005107>)). So, in essence, we adopted the independent filtering for both increasing the number of DRBs and removing the tags with non-NB distribution. To our understanding, this is an elegant solution to prevent possible false discoveries originating from the non-NB reads while increasing the total number of detected DRBs.

Minor comments

Our response: We thank the Reviewer for the careful review of our work and pointing out many useful improvements on the manuscript. We have addressed most of the minor issues raised by the Reviewer, please see below our point-by-point responses

The use of the term 'lineage' more generally refers to a differentiation trajectory of cells; the term 'clones' or 'clonal expansion' might describe the given conditions more precisely.

Our response: We thank the Reviewer for the advice. We have replaced the term 'lineage' with 'clone' or 'clonal-lineage' throughout the manuscript.

The distinction from the experimental design proposed here to single cell sequencing experiments can be made explicit, particularly w.r.t the t-SNE in Fig 5a.

Our response: We have substantially revised the **Multidimensional phenotypic profiling section (p. 9-10)**. We hope the improved text will prevent the reader from confusing the profiling done in our work with scRNA seq experiments.

Other tested DEseq variants; why does edgeR fail like DEseq does?

Our response: We could not, unfortunately, find any other variants for the significance testing in DESeq. To our knowledge, there is only one variant for the statistical testing implemented in the DESeq workflow. We believe that edgeR experience the same problem of improper variance estimation than DESeq.

Fig. 3c: A classical ROC-curve plot annotated with the AUC would provide a better overview of the differences in performance of the individual methods and highlight the benefit provided in DEBRA

Our response: We agree that in some applications ROC-curve is more informative than precision-recall (PR) curves. However, as in our classification problem the true positive class consists of both enriched and depleted barcodes, without having a separate true negative class definition, we feel the PR analyses are better-suited than ROC analyses that require ground-truth definition of both true positive and negative classes. Furthermore, according to previous research [<https://doi.org/10.1371/journal.pone.0118432>], the PR curve is a more suitable option for the comparative analysis of the algorithm performance in imbalanced datasets (here, variable proportions of enriched barcodes). We have better described the reasons for using PR analyses and pAUC metric in the **Methods section (page 20, par 3)**.

The role of Fig 3d and e are unclear in the context. There is no ground truth known and no interpretation as to the observed differences in DRB detection offered, hence these figures might better fit in the supplement.

Our response: We have moved this figure to the appendix (Appendix Fig S7).

It is unclear how the validation of the phenotypic profiling (Fig. 4) relates to the improvement in DRB detection. If there is no tangible connection, the manuscript's cohesion might be improved by combining Figures 4 and 5.

Our response: We have merged the figures as advised by the Reviewer.

The clusters identified in Fig 5a are not clearly defined. Perhaps the authors can repeat the clustering with other projections such as PCA, UMAP, or MDS, either to identify novel clusterings or demonstrate the stability of existing clusters.

Our response: Thank you for the valuable advice. We have repeated the clustering with t-SNE and UMAP (**Fig5 FGH**) which demonstrated the stability of previously identified clusters.

Additionally describe generation of read count tables, what barcode ratios represent, and why DEBRA will be better able to approximate them.

Our response: We have included separate Methods subsection (**page 18**), where we described the generation of the read count tables with different proportions of enriched barcodes (see also Appendix Fig S2).

Regarding the better performance of the DEBRA for the samples with lower proportion of the enriched barcodes, **we have now added a short discussion (page 11, last paragraph; page 12, first paragraph)** that the results from samples with different proportions of enriched barcodes (called barcode ratios in the previous version of the manuscript) should be assessed in comparison to expected “random” FDR values (obtained when p-values were randomly permuted over the barcodes, indicated by black lines in Fig EV3). The apparently low percentage of false discoveries observed for the samples with 0.5 enriched barcodes when analyzed by the original algorithms can be misleading if assessed in isolation from “random” results (black lines). This is because all the barcodes in these samples have significantly perturbed true means, so when the original algorithm produced improper statistical inference, the likelihood of the correct class assignment (enriched or depleted) is high, hence, such a barcode is considered as a correctly called DRB. However, in the dataset with only a fraction of barcodes being differentially represented (0.15 and 0.05 proportions of enriched barcodes), which is more realistic in our opinion, the fold changes of the non-enriched barcodes approach zero. Therefore, mistakenly called DRB has a higher chance to be assigned to a wrong class, and hence, called as a false discovery.

Was any preprocessing/PCA etc of phenotype vectors carried out before running t-SNE?

Lineage read count cutoff of 70 in Fig. 4 vs 75 in Fig. 5.

Our response: We turned off the initial PCA preprocessing step by setting `Rtsne::Rtsne(...,PCA=FALSE,...)`. We now mention this in the Methods section (**page 21, par. 2**).

In the current version, both figures have now a mean read count threshold of 70.

the interpretation of this sentence is unclear: "10⁵ times less of [sic] individual sequencing tags as compared to an RNA-seq sample produced from the same number of cells".

Our response: We agree that in the originally-submitted manuscript, this point was not clearly explained. We have clarified this part in the current version (**page 11, par. 2**)

The Discussion mentions that simulated effect sizes were low, yet it is implied that the experimental setups that the simulated lineage tracing experiment were intended to follow are carried out in the context of "very narrow sample size

bottlenecks", i.e high effect sizes. Are higher perturbation degrees in the simulated experiment required in order to accurately reflect real experimental conditions?

Our response: Thank you for the very relevant comment; indeed in some experimental setups, the effect sizes might be larger than those simulated in the dataset. For example, in the case of mutation driven drug resistance. However, cellular DNA-barcoding experiments do not imply high effect sizes *per se*, even in the case of a very narrow sample size bottlenecks. The effect size depends on clone-specific phenotype, rather than the extent of the selection pressure. For example, if one treats cells with LD99, this imposes a narrow sample bottleneck, but if the clones have no difference in drug response, then the effect sizes are essentially zero.

When designing the experiment for generating a benchmark dataset, we simulated low effect sizes with a purpose to enable testing of the challenging conditions for more thorough comparison between different analysis algorithms, as higher effect sizes are supposed to be easier to detect. We did this as wanted to prevent researchers from calling non-differentially responding clones, as follow-up experiments are costly and time consuming. Using a published dataset (Seth et.al. 2019), we demonstrated how the strong selection pressure could lead to a very high differences in variance, and hence, biased DRB detections with high rate of false discoveries.

We believe that the new set of experiments made for the revised manuscript (**new Fig 4, Fig EV4, and Appendix Fig 9 and the new results section on page 8**) validates the improved performance of the DEBRA algorithm under the real phenotypic pressure.

2nd Editorial Decision

7th February 2020

Thank you again for sending us your revised study. We have now heard back from the two referees who were asked to evaluate your study. As you will see below, the reviewers acknowledge that the performed revisions have addressed the previously raised issues. However, reviewer #3 still lists a couple of remaining concerns, which can be addressed by text modifications, and we would ask you to address in a minor revision.

REFeree REPORTS

Reviewer #2:

The authors have addressed my previous concerns.

Reviewer #3:

The authors have addressed my previous review comments in a satisfactory manner and have improved the clarity of their figures considerably. However, the authors should tame down the claim

in the abstract concerning the significance of the observed improvement in differential barcode detection accuracy and be more specific about the improvement. Figures 3C & D indicate an improvement in the range of 5-10% over existing methods while the vanilla DEseq results appear comparable to DEBRA with shrunken dispersion.

Minor points:

- Fig. 5A x-axis labeled as "lineage" vs "clone" in Fig. 5B
- Missing parenthesis in first paragraph of Discussion

2nd Revision - authors' response

13th February 2020

Reviewer 3:

The authors have addressed my previous review comments in a satisfactory manner and have improved the clarity of their figures considerably. However, the authors should tame down the claim in the abstract concerning the significance of the observed improvement in differential barcode detection accuracy and be more specific about the improvement. Figures 3C & D indicate an improvement in the range of 5-10% over existing methods while the vanilla DEseq results appear comparable to DEBRA with shrunken dispersion.

Our response: We thank the reviewer for the positive comments on our revised manuscript.

Indeed, it can be argued that the DEBRA method provides a moderate improvement in DRB prioritization as seen from the precision-recall curves (Figure 3D) and the corresponding ROC analyses (Figure 3C). However, the major drawback of the traditional methods, such as DESeq, is their overestimation of the significance level, i.e., $-\log(\text{FDR})$, which often leads to a high proportion of false positives. Specifically, as shown in Figure 1D, the traditional methods detect as much as half of the barcodes as DRBs in the samples with no DRBs (null samples) at FDR 25%. Similar behavior is seen for the perturbed samples, where the proportion of incorrectly detected DRBs can reach ~80% (at FDR 25%) in some of the tested conditions. In our opinion, these results make the RNA-seq analysis algorithms inapplicable for accurate analysis of the clone tracing experiments with strong selection pressure.

As requested, we have now toned down and made more specific the statement about the improvements in the revised abstract (page 2).

Minor points:

- Fig. 5A x-axis labeled as "lineage" vs "clone" in Fig. 5B
- Missing parenthesis in first paragraph of Discussion

Our response: We have corrected these as follows:

- The Fig. 5B x-axis is labelled now as "Clone"
- The missing parentheses is added on page 11.

Thank you again for sending us your revised manuscript. We are now satisfied with the modifications made and I am pleased to inform you that your paper has been accepted for publication.

Corresponding Author Name: Tero Aittokallio

Manuscript Number: MSB-19-9195